

# CO₂ and CO temporal variability over Mexico City from ground-based total column and surface measurements

Noémie Taquet[1], Wolfgang Stremme[1], María Eugenia González del Castillo[1], Victor Almanza[1], Alejandro Bezanilla[1], Olivier Laurent[2], Carlos Alberti[3], Frank Hase[3], Michel Ramonet[2], Thomas Lauvaux[4], Ke Che[4], Michel Grutter[1]

[1]Instituto de Ciencias de la Atmósfera y Cambio Climático, Universidad Nacional Autónoma de México, México
[2]Laboratoire des Sciences du Climat et de l'Environnement (LSCE), IPSL, CEA-CNRS-UVSQ, Université Paris-Saclay, Gif-sur-Yvette, France
[3]Institute of Meteorology and Climate Research (IMK-ASF), Karlsruhe Institute of Technology (KIT), Karlsruhe, Germany
[4]Groupe de Spectrométrie Moléculaire et Atmosphérique (GSMA), Université de Reims-Champagne Ardenne, UMR CNRS 7331, Reims, France

*Correspondence to*: Noémie Taquet (noemi.taquet@gmail.com)

**Abstract.** Precise estimates of greenhouse gas (GHG) emissions and sinks are critical for understanding the carbon cycle and identifying key factors in the human-induced climate change feedback. Recent efforts were focused on reconciling bottom-up and top-down GHG emissions estimates, in particular on the city scale, using both space-based and ground-based atmospheric composition measurements that still show serious discrepancies. In this study, we explore the variability of the CO and $CO_2$ emissions from the Mexico City Metropolitan Area (MCMA) from long-term time-resolved total column measurements using solar absorption Fourier Transform Infrared Spectroscopy (FTIR). Measurements were performed at three stations, two of them located in the urban area at two opposite sides of Mexico City center and the third in a mountainous background site. Using a simple model and the mixed layer height from a ceilometer, the GHG concentration in the mixed layer and the $CO/CO_2$ ratio were determined from the total column observations and compared to surface measurements using Cavity ring-down spectroscopy (CRDS). Finally, combining the ground-based total column and space-based TROPOMI CO measurements, we estimate the annual CO and $CO_2$ MCMA emissions based on a simple model, i.e.: without recourse to complex transport models. By this way, we study the inter-annual variability of the CO and $CO_2$ MCMA anthropogenic emissions, and relate it to the main natural or anthropogenic changes occurring during the last decade, such as the 2015-2016 El Niño period or the COVID-19 lock-down event.

## 1 Introduction

The greenhouse gas (GHG) mitigation strategies implemented in megacities following the 1997 Kyoto Protocol and the 2015 Paris Agreement play a crucial role in the global action plan to mitigate climate change, given that cities are accountable for more than 70% of the global anthropogenic emissions (Duren and Miller, 2012). With the recent progress in space-based and ground-based remote GHG measurements in terms of accuracy, spatial coverage/resolution and temporal frequency, GHG emissions can increasingly be constrained by comparing bottom-up and top-down estimates. Top-down approaches are generally based on ground or space-based atmospheric measurements coupled with inverse modelling, using 3D-Eulerian (i.e: WRF-Chem) or Lagrangian and hybrid (i.e: STILT, Hysplit) approaches (Wu et al., 2018, Che et al., 2022; Lian et al., 2023). The quantification of anthropogenic $CO_2$ enhancements from cities using satellite data e.g: GOSAT (Wang et al., 2019), OCO-2 (Ye



et al., 2020) or TanSat (Liu et al., 2018) is still challenging due to the sparsity of the observations, the low signal
from the anthropogenic contribution compared to the background levels and biogenic contribution, and some
inconveniences inherent to space-measurements such as the non-negligible aerosol effects (Wang et al., 2020 and
references therein). Some studies have estimated the urban enhancements of anthropogenic $CO_2$ concentrations
along with CO and $NO_2$ from satellite measurements, as these air pollutants can serve as tracers of anthropogenic
$CO_2$ (Silva et al.,2013; Park et al., 2021 and references therein). The $CO/CO_2$ ratio is often used to determine the
combustion efficiency of the cities (Park et al., 2021 and references therein). With the development of a new
generation of space-based observatories, such as Sentinel-5P and OCO-2,3, the evolution of GHGs at the city scale
can now be characterised with a finer temporal and spatial resolution (Kiel et al., 2021) but more validation efforts
are needed. As inverse modelling is likely undermined by the approximations used for defining the emission
patterns, transport processes and meteorology, top-down approaches may lead to discrepancies in emissions
estimates, in particular in sites with complex orography.
Ground-based total column FTIR instruments provide valuable long-time concentration measurements of
GHG and pollutant reactive species, as well as anthropogenic tracers, constituting a key element to validate
regional and local inventories. Some studies  reported estimates of $CO_2$ and $CH_4$ emissions from large urban areas
(Babenhauserheide et al., 2018 in Tokyo; Hedelius et al., 2018 in the California Southern Coast Air Basin
California megacity), using data from high-resolution FTIR instruments (i.e: Bruker IFS120/5HR) contributing to
the Total Column Carbon Observing Network (TCCON). Nevertheless, only a few TCCON stations are located in
urban areas (Toon et al., 2009; Chevallier et al., 2011; Sussman et al., 2020). The development of the COllaborative
Carbon Column Observing Network (COCCON, Frey et al., 2019), using a new generation of portable low spectral
resolution FTIR spectrometers (EM27/SUN, Gisi et al., 2012; Hase et al., 2016) able to simultaneously measure
the $CO_2$, CO, $H_2O$ and $CH_4$ average total columns with a similar quality as TCCON, has considerably densified
the number of measurements in urban environments. Some studies reported emission estimates for big cities by
means of the deployment of several EM27/SUN instruments at strategic sites throughout the cities (Hase et al.,
2015 and Zhao et. al., 2019 in Berlin; Vogel et al., 2019 in Paris; Makarova et al., 2021 in St Petersbourg; Zhou et
al., 2022 in Beijing and Xianghe; Che et al. 2022, in Beijing; Rißmann et al., 2022 for Munich) coupling
measurements with inverse modelling. Most of these studies were based on short-term campaign observations,
applying the Differential Column Methodology (DCM, Chen et al., 2016) or dedicated dispersion models (Hase
et al., 2016), coupled with simple mass balance-based methods or inverse modelling to derive emissions. Most of
these studies reported significant discrepancies between the estimates, depending on the models used (Viatte et
al., 2017).
In this study, we aimed to determine the Mexico City Metropolitan Area (MCMA) $CO_2$ and CO emissions
using ground-based FTIR and surface measurements, without resorting to complex dispersion and/or chemistry
transport models. The MCMA, with a population around 22 million inhabitants, is in the top ten most populous
cities in the world and ranks among the major emitters of GHGs in North America. The available information of
GHGs emission estimates are mainly based on the inventories reported by the Ministry of the Environment of
Mexico City (SEDEMA), which is updated every two years, but lagging several years behind. In the report based
on 2018, the latest published before the COVID19-lock-down (2020), a total emission of 75.2 Mt CO2-eq is
estimated for the MCMA, 87% of which is attributed to fossil fuel combustion and 58% originates from the
transport sector (SEDEMA Inventory, 2018). The Mexico City government is actively engaged in the C40 Climate



Change Program and implemented significant policy measures since 2008, including promoting sustainable
transportation systems, implementing energy efficiency measures, increasing the use of renewable energy sources,
and adopting green building practices. On a national scale, the country is committed to reduce its GHGs emissions
by 35% by 2030 with respect to its base level, as stated in the last Nationally Determined Contributions report
(NDC-2022, UNFCCC). To assess the effect of the national and local mitigation policies, the installation of
ground-based GHG measurement networks and the refinement of bottom-up estimates by comparing them with
the top-down method (i.e: inverse modelling) is of critical importance to obtain a comprehensive GHGs database
that can serve as follow-up of the mitigation actions.
The Institute of Atmospheric Sciences and Climate Change (ICAyCC, Spanish acronym) at UNAM
(Universidad Nacional Autónoma de México) deployed in the last decade a wide range of surface gas sensors and
ground-based remote sensing instruments across the MCMA (Grutter, et al., 2003; Molina et al., 2010; Bezanilla
et al., 2014; Stremme et al., 2009; 2013; Baylon et al., 2017) in the frame of research projects related to air quality
assessment and validation of ground-based satellite products. Since 2013, UNAM has contributed to the Network
for the Detection of Atmospheric Composition Change (NDACC), performing continuous composition
measurements of the free troposphere from the high altitude Altzomoni Atmospheric Observatory (ALTZ) station,
located 60 km southeast of Mexico City at 3985 m a.s.l. Baylon et al., (2017) reported the background $CO_2$
variability and trend from this station between 2013 and 2016. Stremme et al., (2013) reported the first top-down
estimate of carbon monoxide (CO) emissions for the MCMA, based on FTIR CO total column measurements and
the Infrared Atmospheric Sounding Interferometer (IASI) data. These authors derived the $CO_2$ emissions for the
MCMA using the CO emission estimates and the average $CO/CO_2$ ratio reported in Grutter (2003), using FTIR
measurements. In 2018, the Mexican/French "Mexico City's Regional Carbon Impacts (MERCI-CO2)" project
(coordinated by UNAM and LSCE) was launched aiming to assess the $CO_2$ emissions from MCMA using
EM27/SUN measurements and inverse modelling to evaluate the effectiveness of the mitigation strategies
implemented by the local authorities. Xu et al., (submitted) examined the performance of a modelling system based
on WRF-Chem to assess the whole-city emissions using the EM27/SUN measurements deployed in the frame of
the MERCI-CO2 project. The complex orography of the region posed a challenge in the atmospheric transport
simulations and thus for the top-down estimates using inverse modelling. Indeed, Mexico City is situated in a high
altitude basin (~2300 m. a.s.l.), surrounded by mountains reaching up to 5.6 km a.s.l., and is prone to accumulate
anthropogenic emissions, especially during the dry season, when the atmospheric boundary layer ventilation is
limited (Burgos-Cuevas et al., 2023). The boundary layer dynamics in the basin and the wind surface circulation
is complex, due to the temperature contrasts and rough topography.
In this study, we report the long-term (2013-2021) variability of the $CO_2$ and CO total columns and
surface concentrations (from 2014) above the MCMA using long-term ground-based FTIR and surface Cavity
Ring-Down Spectroscopic (CRDS) measurements. Using the mixed layer height data from the continuous
ceilometer measurements at UNAM, we examined the consistency of the surface and total column measurements
of our network. We also determined an average $CO/CO_2$ ratio based on FTIR and surface measurements at different
temporal resolutions (from daily to intraday). Then, using the spatial distribution of TROPOMI CO column
measurements, we explore the potential of our FTIR network to capture the variability of the megacity CO and





$CO_2$ emissions using a simplified model, i.e.: without recourse to complex numerical simulations. Our estimates
are compared with the available bottom-up and previous top-down estimates.

**2 Sites, instrumentation and measurement protocols**

We used in this study the column-averaged dry-air mole fractions of $CO_2$ and CO (XCO2 and XCO) from
three permanent FTIR stations distributed in a radius of 100 km around MCMA (Fig. 1), and the surface
measurements performed at UNA and ALTZ sites. The measurement periods for the different instruments at each
site are reported in Table 1. The VAL station is located at the northern part of the city in a highly industrialised
zone. The UNA station is situated at the south of the city in the main campus of UNAM. The third station is the
ALTZ background site (3985 m a.s.l.), located 60 km ESE from UNAM, within the Izta-Popo National Park. The
equipment of the different stations and measurement protocols are described in the following sub-sections.

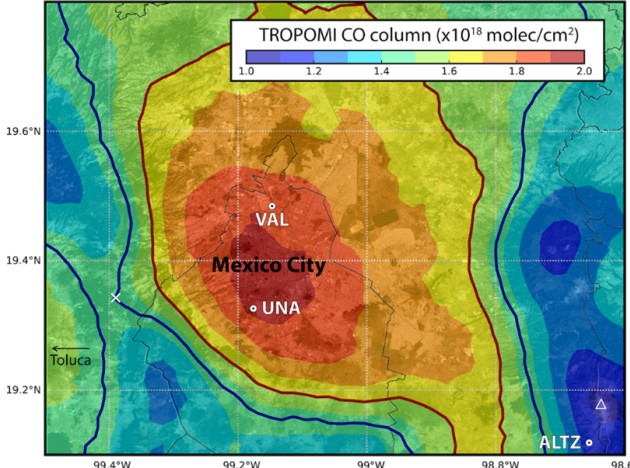

**Figure 1: Map of the ALTZ, UNA and VAL stations and average distribution (2018-2022) of carbon monoxide total**
**columns over the Mexico City Metropolitan Area (MCMA) calculated from the TROPOMI CO product. The average**
**total column can be decomposed into two main contributors: i) a background of around $1.45 \times 10^{18}$ molec.cm$^{-2}$ (limits**
**represented by blue contour lines) and ii) the local influence corresponding to the carbon monoxide emitted on the same**
**day. The total columns are highly influenced by the topography which is clearly visible over the highest terrains of the**
**region, near the Volcanoes Popocatepétl and Iztaccíhuatl (close to the ALTZ station) at the south east of Mexico City.**
**The mountains of Ajusco are located southwest of Mexico City. The enhancement in the center of the metropolitan area**
**reflects the carbon monoxide locally emitted on the same day.**
*Table1: Instrumentation and measurement periods used in this study.*

| Station | Instrument | Measurement period | Product |
|---------|------------|--------------------|---------|
| ALTZ | IFS120/5HR | 01/01/2013 - 01/06/2021 | XCO and XCO2 |
| | EM27/SUN #038 | 21/10/2020 - 20/12/2020 & 10/02/2021 - 22/02/2021 | XCO and XCO2 |
| | EM27/SUN #104 | 07/02/2020 - 18/02/2020 | XCO and XCO2 |
| | CRDS G2401 Picarro | 15/11/2015 - 01/06/2021 | Surface CO and CO2 |



| UNA | Vertex | 15/11/2015 - 20/06/2017 | XCO |
|-----|--------|------------------------|-----|
|  | EM27/SUN #038 | 07/05/2021 - 25/05/2021 | XCO and $XCO_2$ |
|  | EM27/SUN #062 | 17/03/2016 - 01/06/2017 | $XCO_2$ |
|  |  | 01/06/2017 - 01/06/2021 | XCO and $XCO_2$ |
|  | EM27/SUN #104 | 04/04/2019 - 19/09/2019 | XCO and $XCO_2$ |
|  | CDRS G2401 Picarro | 15/11/2015 - 01/06/2021 | Surface CO and $CO_2$ |
|  | CL31 Vaisala ceilometer | 15/11/2015 - 01/06/2021 | Mixed Layer Height |
| VAL | EM27/SUN #104 | 23/09/2019 - 01/06/2021 | XCO and $XCO_2$ |


## 2.1 The UNA station: Total columns, surface concentrations and mixed-layer height measurements


147       Atmospheric total columns of several gas species, such as $O_3$, $NH_3$, $CH_4$, CO, and HCHO have

continuously been measured at UNA since 2010 (Bezanilla et al., 2014; Plaza-medina et al., 2017; Baylon et al.,
2017; Rivera-Cardenas et al., 2021; Herrera et al., 2022) using solar absorption FTIR spectroscopy.
Measurements are performed in the mid-infrared (MIR) and near-infrared (NIR) spectral ranges using a Bruker
model Vertex 80 spectrometer. The instrument has a Maximum optical Path Difference (MPD) of 12 cm
(corresponding to a spectral resolution of 0.075 cm$^{-1}$) and is equipped with two detectors, a liquid-nitrogen cooled
mercury-cadmium-telluride (MCT) and InGaAs detectors. Solar absorption measurements are performed using a
home-built solar tracker. A full description of the instrumental set-up and measurement protocols is given in
Bezanilla et al. (2014) and Plaza-Medina et al. (2017). The CO measurements are routinely performed in the MIR
spectral range with a spectral resolution of 0.1 cm$^{-1}$, using the MCT detector.

157       In March 2016, an EM27/SUN spectrometer was implemented at UNA to continuously measure $XCO_2$,

$XCH_4$, $XH_2O$, XCO total columns from solar NIR spectra with a spectral-resolution of 0.5 cm$^{-1}$ (MPD of 1.8 cm).
The spectrometer is equipped with its own solar tracker (Bruker CAMTracker; Gisi et al., 2011) capturing and
redirecting the solar beam into a RockSolid$^{TM}$ pendulum interferometer equipped with a Quartz beamsplitter. The
EM27/SUN, with serial number #62 installed at the UNA station (hereafter EM27-SUN_62), was initially operated
with a standard InGaAs-diode detector sensitive to the 5500-11000 cm$^{-1}$ spectral range, to which a second InGaAs
detector with Ge filter was added in 2017 for CO measurements through a second channel (4000 – 5500 cm$^{-1}$)
(Hase et al., 2016). Further details on the technical characteristics and systematic performance evaluation of the
EM27/SUN spectrometer are given in Frey et al., (2019) and Alberti et al., (2022). The spectrometer was installed
in a home-made protective box, including a remotely-controlled dome cover, a GPS and a PCE-THB-40 data-
logger for precise timing and surface pressure measurements. Double sided forward-backward interferograms are
routinely recorded with a scanner velocity of 10 kHz, so that the recording time of one measurement (averaging
10 IFGs scans) is close to one minute.

170       Additionally, $CO_2$, CO, $CH_4$ and $H_2O$ surface measurements are continuously performed at the UNA

station using a Cavity Ring-Down Spectrometer (CRDS, model G2401 from Picarro Inc.). The CRDS spectrometer
uses a laser to quantify the spectral features of gas-phase molecules in an optical cavity offering effectively of up
to 20 km absorption path length. Frequency shifts are prevented with a high-precision-wavelength monitor and
temperature and pressure are precisely controlled by the analyzer. The quantification is improved by the
simultaneous spectral analysis of the measured gases. A calibration system using 3 gas standards provided by the
National Oceanic and Atmospheric Administration Earth System Research Laboratory (NOAA ESRL), traceable



to the WMO2007 scale, was set up in 2018 at UNA and in 2019 at ALTZ. Data collected before the installation of
the calibration systems were corrected with calibration coefficients obtained in 2018. The sampling inlet using
Synflex tubing was placed at 24 m a.g.l. at UNA station and includes a Nation air dryer, as described in detail by
González del Castillo et al. (2022). Data are continuously collected at 0.3 Hz rate and their uncertainties, calculated
as the standard deviation of raw data over 1-minute intervals when measuring calibration gases, are equal to 0.03
ppm at UNA (González del Castillo et al., 2022).

Finally, continuous mixed-layer height (MLH) measurements are performed since 2008 at UNA using a
CL31 ceilometer instrument (Vaisala). This is a robust commercial instrument which emits light pulses at 10 kHz
repeating frequency at 910 nm using an indium-gallium-arsenide diode laser. It detects the backscatters signal
through a single lens with a silicon avalanche photodiode. The resulting backscattering profiles have a vertical
resolution of 10 m and reach an altitude of 7,500 m. The profiles have been used to retrieve MLH above the city
since 2011 (García-Franco et al., 2018).

**2.2 The ALTZ background station: Total columns and surface measurements**

The Altzomoni Atmospheric Observatory (ALTZ) was equipped with a high-resolution FTIR
spectrometer (model IFS120/5HR, Bruker) in 2012, capable of measuring atmospheric spectra in the NIR and MIR
spectral regions with 257 cm MPD, equivalent to a spectral resolution of 0.0035 cm$^{-1}$. The instrument is installed
into a container with a motorised dome cover on the roof and a microwave communication system (60 km line-of-
sight to the university campus), which allows a fully-remote control of the instruments. When the dome is open, a
solar tracker (CAMTracker; Gisi et al., 2012) collects the solar beam and orients it toward the spectrometer
entrance. The spectrometer can be operated with KBr or CaF$_2$ beam splitters, 3 different detectors (MCT, InSb,
and InGaAs) and a set of 7 optical filters is installed in a rotating wheel. The measurement routine consists in the
acquisition of high (0.005 cm$^{-1}$), medium (0.02 cm$^{-1}$ and 0.1 cm$^{-1}$) and low (0.5 cm$^{-1}$) resolution spectra in the NIR
and MIR spectral ranges using the different NDACC filters (~40 min for a complete sequence).
The NIR CO and CO$_2$ spectra (0.02 cm$^{-1}$) used in this study were recorded as the average of two scans
taken for approximately 38 s with a scanner speed of 40 kHz. The MIR CO spectra (0.005 cm$^{-1}$) are deduced from
the coaddition of 6 scans (<200 s) with a scanner speed of 40 kHz. Due to a spectrometer laser replacement, the
IFS120/5HR measurements were interrupted between November 2020 and February 2021 (Table 1). To avoid an
important gap in the measurements, an EM27/SUN (EM27/SUN_38) was temporarily installed at the station
during this period. The intercalibration factors used for combining the two types of measurements were determined
from previous side-by-side measurements performed during February 2021 (see Table S1 and section 3.1.3).
A CRDS (model G2401 from Picarro Inc.) instrument was implemented at the station in 2014 providing
continuous CO$_2$, CO, CH$_4$ and H$_2$O surface measurements (Gonzáles del Castillo et al., 2022).  The sampling inlet
using Synflex tubing was placed at 4 m a.g.l. and includes a Nation air dryer (similar installation to UNA). A
calibration system similar to that implemented at UNA, using 3 NOAA ESRL gas standards, was set up in 2019.
The station also includes meteorological instruments, pressure and temperature sensors and visible cameras among
other instrumentation for atmospheric and environmental monitoring.




**2.3 The VAL station: Total column measurements**


215         The VAL station, located in Vallejo in the northern part of MCMA, is part of the city's air quality network

(RAMA) run by SEDEMA. An EM27/SUN spectrometer (EM27/SUN_104) was installed at this station in 2019
together with a surface $CO_2$ sensor. The VAL spectrometer has been performing measurements with the two
detectors since November 2019.
**3 Data Analysis**
**3.1 FTIR data processing and analysis**

221         In this study, we used the solar absorption measurements acquired by five different FTIR instruments

(i.e: three EM27/SUN, a Vertex 80 and a IFS120/5HR) to estimate the $XCO_2$, and XCO total columns at each
station. The retrieval strategies were adapted as a function of the spectral resolution and averaging kernel of each
species. Table 2 summarises the different products used in this study, and their retrieval parameters.

*Table 2: FTIR analysis: Description of the different FTIR products, retrieval strategies and parameters used in this study.*

| Instrument (spectral resolution) | Gas | Microwindows (cm$^{-1}$) | Interfering gases | Retrieval code | Retrieval method |
|---|---|---|---|---|---|
| EM27/SUN and IFS-120/5HR LowRes (0.5 cm$^{-1}$) | $CO_2$ | 6173.0-6390.0 | $H_2O$, $CH_4$ | PROFFAST | Scaling VMR COCCON strategy |
| | CO | 4208.7-4318.8 | $H_2O$, HDO, $CH_4$, HF | | |
| | $O_2$ | 7765.0 - 8005.0 | $H_2O$, $CO_2$, HF | | |
| IFS-120/5HR (0.02 cm$^{-1}$) (TCCON-type) | $CO_2$ | 6180.0 – 6260.0 6310.0-6380.0 | $H_2O$, $CH_4$,HDO | PROFFIT9.6 | Scaling VMR |
| | CO | 4208.7- 4257.3 4262.0 – 4318.8 | $CH_4$, $H_2O$, HDO | | |
| | $O_2$ | 7765.0-8005.0 | $H_2O$, $CO_2$ , HF | | |
| IFS-120/5HR (0.005 cm$^{-1}$) (NDACC-type) | CO | 2057.70-2058.00 2069.56-2069.76 2157.50-2159.15 | $O_3$, $N_2O$, $H_2O$, OCS and $CO_2$ | PROFFIT9.6 | Profile NDACC strategy |
| Vertex80 (0.1cm$^{-1}$) | CO | 2056.70 – 2059.00 2068.56-2069.77 2156.50-2160.15 | $O_3$, $N_2O$, $H_2O$, OCS and $CO_2$ | PROFFIT9.6 | Profile |


**3.1.1 EM27/SUN spectra analysis**
Double-sided interferograms from the EM27/SUN were analysed following the standardised COCCON protocol,
using PREPROCESS and PROFFAST codes, developed by the KIT and made freely available (https://www.imk-
asf.kit.edu/english/COCCON.php). The codes and retrieval methods are fully described in Sha et al. (2020), Frey
et al. (2021) and Alberti (2023) and only briefly summarised here. The PREPROCESS algorithm generates the



required spectra by a Fast Fourier Transform. The processing incorporates various quality checks, as a signal
threshold, intensity variations during recording, requirement of proper spectral abscissa scaling, and generates
spectra only from raw measurements passing all checks (the remaining ones being flagged). We used the ILS
parameters (i.e: modulation efficiency amplitude and phase error) reported on the KIT-COCCON website
(https://www.imk-asf.kit.edu/english/COCCON.php) and in Alberti et al. (2022), corresponding to the initial KIT
calibration of the spectrometers (Frey et al., 2019, Alberti et al., 2022). The PROFFAST-PCXS module (i.e:
forward model of PROFFAST) pre-calculates daily lookup tables of the molecular absorption cross-sections
according to the meteorological parameters and gas trace VMR profiles priors. The latest PROFFAST-PCXS
version uses the HITRAN 2020 spectroscopic linelists (with some extensions, e.g., line mixing parameters added
for CH4). Here, we used the standard COCCON linelists as incorporated in the previous PROFFAST version, i.e:
HITRAN 2008 for $CH_4$, HITRAN 2012 for $CO_2$, a modified version of HITRAN 2009 by Toon (2014) for $H_2O$, a
TCCON standard linelist for $O_2$, and the same solar line list as previously used by TCCON (compiled by G.C.
Toon for GGG2014). The least-squares fitting code PROFFAST-INVERS retrieves the total columns by scaling
the prior VMR profiles iteratively until adjusting the fit to the measured spectra. The intraday variability of surface
pressure is considered in the retrieval, interpolated from the in-situ pressure measurements. For tying the column-
averaged abundances provided by COCCON to TCCON data, PROFFAST applies post-process Airmass-
Dependent (ADCF) and -Independent (AICF) corrections, independent from the instrument, similar as used in the
TCCON process (Sha et al., 2020, and Alberti, 2023). The corrections and parameters used are reported in the
COCCON website and Alberti, (2023).
We automatized and adapted the data processing to obtain a preliminary "real-time" hourly-updated
analysis (hereafter, AN1) for each site, additionally to the off-line treatment (hereafter, AN2) applying the standard
COCCON procedure. The meteorological data used in the AN1 retrieval were derived from the daily-available
radiosonde data, provided by Servicio Meteorologico Nacional (SMN) from measurements performed in the early
morning (6 AM LT) at the Mexico City International Airport. The AN1 strategy adopted fixed VMR priors for
each species, consisting in the averaged profile of 41 years (1980-2020) run of the Whole Atmospheric Community
Climate Model (WACCM), as commonly used in the NDACC community. The AN2 processing, generating the
COCCON standard products, used the daily TCCON meteorological data and priors (GGG2014 version of MAPs
files), downloaded from the Caltech server, which are based on National Centers for Environmental Prediction
(NCEP) reanalysis. For both AN1 and AN2 processing, we used the in situ intraday surface pressure measurements
from the PCE-THB-40 sensors. A correction factor was applied to the pressure measurements to take into account
the bias between the different pressure sensors used, previously intercompared by a few days of side-by-side
measurements.
$CO_2$, $O_2$, and CO were analysed in the 6173.0-6390.0 cm$^{-1}$, 7765.0- 8005.0 cm$^{-1}$ and 4208.7- 4318.8 cm$^{-1}$
spectral windows, respectively. The XCO$_2$ and XCO column-averaged dry air mole fractions were calculated using
the $O_2$ retrieved total columns, according to Wunch et al. (2009):
$Xgas = 0.2095 \ (C_{gas} \ / \ C_{O2})$                                     (1)
where C$_{gas}$ and C$_{O2}$ are the target gas and $O_2$ total columns, respectively.
The real-time (AN1) and COCCON (AN2) XCO$_2$ and XCO products showed relative differences lower than 0.05%
and 5%, respectively. The results presented hereafter are based on the official COCCON products (AN2 analysis).



**3.1.2 Vertex80 and IFS120/5HR spectra analysis**

High (0.005 cm$^{-1}$) and medium (0.02 cm$^{-1}$ and 0.1 cm$^{-1}$) resolution solar-absorption spectra are processed using the PROFFIT9.6 code (Hase et al., 2004).

XCO$_2$ is retrieved from the NIR 0.02 cm$^{-1}$ resolution spectra applying the procedure described in Baylon et al. (2017), in which two independent CO$_2$ and O$_2$ VRM-scaling retrievals are performed using fixed WCCAM VMR priors and NCEP-derived meteorological data. Spectral windows and interfering gases (Table 2) are similar to those used in the standard TCCON procedure. XCO$_2$ is then calculated from the retrieved CO$_2$ and O$_2$ total columns by applying the Eq. (1).

For the ALTZ analysis, CO was retrieved from the high (0.005 cm$^{-1}$) resolution spectra in the MIR region, applying the standard NDACC procedure (Table 2). It uses a profile retrieval strategy with fixed WACCM VMR priors and NCEP meteorological data. Since the O$_2$ specie is not analysed in the MIR region, the XCO was determined using the dry air columns ($C_{dryair}$):

$$XCO = \frac{C_{CO}}{C_{dryair}} \tag{2}$$

with:

$$C_{dryair} = \left(\frac{P_g}{g} \cdot m_{dryair}\right) - \left(C_{H2O}\frac{m_{H2O}}{m_{dryair}}\right) \tag{3}$$

where C$_{CO}$ and C$_{H2O}$ are the retrieved CO and H$_2$O total columns, $g$ the column-averaged gravity acceleration, P$_g$ the ground pressure and m$_{dryair}$ and m$_{H2O}$, the dry air and H$_2$O molecular masses respectively. In addition, we analysed XCO from the NIR spectral region to complement the MIR time-series, occasionally interrupted when the liquid nitrogen was missing at the station. The CO and O$_2$ columns in the NIR region were analysed using scaling retrievals in the same spectral windows as that used by TCCON (Table 2), but with fixed WACCM VMR priors and NCEP meteorological data. XCO was calculated from the CO and O$_2$ retrieved total columns applying the Eq. (1). To minimise the air mass dependence effect (likely low for CO), we filtered out data with a SZA >60°. XCO NIR and MIR products were compared and intercalibrated (section 3.1.3).

For UNA, we used the XCO total columns calculated from the Vertex80 measurements to complement the EM27/SUN time series during the period when it was operating with a single detector (between March 2016 and September 2017). CO was analysed from the 0.1 cm$^{-1}$ resolution spectra in the MIR spectral range, using a standard NDACC profile retrieval strategy and the PROFFIT9.6 retrieval program with constant WACCM VMR priors and NCEP meteorological data. Spectral windows (Table 2) were adapted following Pougatchev and Rinsland (1995). Previous CO total columns time series retrieved from the same method at UNA were presented in Garcia-Franco et al. (2018) and Borsdorff et al. (2018, 2020). Only the constraint of these CO retrievals were adjusted for the Megacity and allowed in addition a free fitting of the mixing layer concentration, following the work by Stremme et al. (2009) in which low resolution MIR- spectra with a different retrieval program have been analysed.

**3.1.3 Measurement precision and FTIR product intercomparison**

Side-by-side measurements were performed at the ALTZ and UNA stations on several occasions (Table1) to assess the FTIR measurement precisions, to characterise the bias between the different products and to define



the inter-calibration factors for the $XCO_2$ and XCO products. We used the EM27/SUN_62 products as reference,
for which we previously applied the standard $XCO_2$ and XCO calibration factors reported in Alberti et al. (2022),
to inter-calibrate our results with the COCCON network and the Karlsruhe TCCON station operated by KIT. The
linear regression parameters from the different measurement pairs and the calibration factors are presented in the
Supplementary data (Table S1 and S2).
We found a bias lower than 0.2% and 1.0% between the three EM27/SUN, for $XCO_2$ and XCO respectively, and
a coefficient of determination ($R^2$) higher than 0.99.
On the other hand, the precision of the EM27/SUN measurements was assessed by calculating the standard
deviation over a 5 min-interval period, and found to be on average 2.7 ppb and 0.3 ppm for XCO and $XCO_2$,
respectively.
The intercomparison of the IFS120/5-HR high resolution (0.02 cm$^{-1}$) products and the EM27/SUN $XCO_2$ products
was performed for the daily average data used in this study. The calibration factors were determined using i) the
EM27/SUN $XCO_2$ products and ii) the IFS 120/5-HR low resolution (0.5 cm$^{-1}$) product (Fig. S2), processed in the
same way as the COCCON EM27/SUN data but having the advantage of being measured even outside the
campaigns carried out with the EM27/Sun. We finally found a bias around 0.4% (slope=0.996), and a coefficient
of determination $R^2$ of 0.92. This bias is in order of that expected when comparing TCCON and COCCON products
(Frey et al., 2019), when no empirical calibration is applied. On the other hand, a bias of 2% (and $R^2$=0.92) was
found comparing the XCO from the EM27/SUN and the Vertex (MIR) products at UNA.
One of the main contributions of the apparent bias observed when comparing products from different instruments
and using different retrieval strategies can be due to their respective Averaging Kernel (AK) which characterise
the smoothing error. It is especially the case in the comparison of XCO from the EM27/SUN (i.e: NIR scaling
retrieval product, Degree Of Freedom (DOF) =1) and from the Vertex (MIR profile-product, DOF > 2). To assess
this effect, we refined the comparison after smoothing the vertically resolved Vertex profiles with the EM27/SUN
AK (following Rodgers, 2000; Borsdorff et al., 2014, 2018) and re-calculating the smoothed Vertex total columns.
After this smoothing, the bias is reduced to 0.2% instead of 4.1% for the CO total columns. For the XCO product,
which includes the use of the surface pressure for the MIR product and the retrieved $O_2$ column for the NIR product
the bias is reduced to 0.4% instead of 3.5%.

### 3.2 Surface CRDS data analysis

The surface $CO_2$ and CO data acquired with the CRDS analysers were processed and averaged following
the procedure described in González del Castillo et al. (2022). Data were averaged and their standard deviation
calculated, per minute, then per hour. To extract the trend and seasonal CO and $CO_2$ variability, data were filtered
by discarding hours generally affected by transient and very local effects. Data recorded between 13 and 17h with
standard deviations lower than 6.0 ppm were selected for the UNA station, while nighttime data (19 to 5h) with
standard deviations lower than 2.0 ppm were selected for the ALTZ station, according to González del Castillo et
al. (2022).

### 3.3 Mixed Layer height from the Lidar measurements

The MLH is retrieved using a combined algorithm based on the gradient method and a wavelet-covariance
transformation as described in detail by García-Franco et al. (2018). These results were compared with radiosonde



data and MLH values derived from surface and vertical column densities of trace gases, and more recently Burgos-
Cuevas et al. (2022) used the variance of the vertical velocity from a Doppler Lidar (Wind Cube 100, Leosphere)
and compared with the ceilometer results at the same location. These studies show that the ceilometer retrieved
MLHs compare well with other techniques during the daytime (they agree within 15% with the trace gas method),
which are relevant for this study, whereas late afternoon and nighttime retrieved values might be affected by
aerosol residual layers at higher altitudes.
**3.4 Mixed layer CO and $CO_2$ concentrations from FTIR measurements**
Pollutant concentrations within the mixed layer are often estimated using surface measurements, although
surface concentrations are very sensitive to the airmass vertical transport, unlike the total columns. It is especially
the case within the Mexico City basin where the mixed layer has a strong diurnal dynamics controlling the vertical
distribution of the emitted pollutants (Stremme et al., 2009; Garcia-Franco et al., 2018). An estimate of the $CO_2$
and CO vertically averaged concentrations across the mixed layer can be made using the total columns measured
at the UNA and ALTZ stations. The dry air mole fraction measured at the UNA station ($XCO_2^{UNA}$) is the weighted
mean of that measured in the mixed layer ($CO_2^{ML}$) and in the free troposphere at the ALTZ station ($XCO_2^{ALTZ}$):

$$XCO_2^{UNA} = w_1 \times CO_2^{ML} + w_2 \times XCO_2^{ALTZ}$$

(4)

$$CO_2^{ML} = \frac{XCO_2^{UNA} - w2 \times XCO_2^{ALTZ}}{W1}$$

(5)

The weights (w1 and w2) depend on the pressure difference between the mixed-layer height (MLH) and the UNA
station, the pressure on top of the mixed layer is calculated assuming an exponential decay and an effective scale
height $H_{scale}$ (assumed to be 8.0 km):

$$w_1 = \left(1 - e^{-\frac{MLH}{Hscale}}\right) \ and \ w_2 = \left(e^{-\frac{MLH}{Hscale}}\right)$$

(6)

The MLH above Mexico City was estimated using the hourly-averaged measurements of the ceilometer
at the UNA station. The hourly-averaged $CO_2^{ML}$ and $CO^{ML}$ products were calculated applying the same strategy
for the entire time series and are reported in Fig. 7, concurrently to the surface data.
**4 Results**
The FTIR $XCO_2$ and XCO daily-averaged time series and $CO_2$ and CO surface concentrations obtained at the
UNA, VAL and ALTZ stations between November 2015 and June 2021 are shown in Fig. 2. Trends and seasonal
variabilities were fitted using a Fourier series analysis (Eq. (7) and black and red solid lines in Fig. 2), following
Wunch et al. (2013):

$$f(x) = ax + \sum_{k=0}^{n} a_k \cos(2\pi kx) + b_k \sin(2\pi kx), \text{ with n} = 2$$

(7)




where $x$ is the time (decimal year), $a$ the mean growth rate (ppm/year), and $a_k$ and $b_k$ the Fourier coefficients
modulating the annual cycles. The coefficients for each gas species and station are reported in Table 3.

*Table 3: Fourier series fitting parameters for the UNA, VAL and ALTZ $XCO_2$ and XCO time series presented in Fig. 2, and*
*calculated from Eq.(7).*

| Fitting parameters (ppm/year) | $XCO_2$, UNA Tot. Col. | $XCO_2$ ALTZ Tot. Col. | $CO_2$ UNA Surface | $CO_2$ ALTZ Surface | XCO UNA Tot. Col. | CO UNA Surface |
|---|---|---|---|---|---|---|
| $a$ | 2.25±0.02 | 2.40±0.01 | 1.6±0.1 | 2.48±0.02 | $(-4.0±0.8)×10^{-3}$ | $(-2.7±0.1)×10^{-2}$ |
| $a1$ | -1.06±0.04 | -0.78±0.04 | 1.7±0.2 | -0.39±0.05 | $(-2.4±0.7)×10^{-3}$ | $(6.5±0.4)×10^{-2}$ |
| $a2$ | 2.11±0.04 | 1.93±0.04 | 1.1±0.2 | -0.36±0.05 | $(-3.2±0.8)×10^{-3}$ | $(1.5±0.4)×10^{-2}$ |
| b1 | 0.71±0.04 | 0.64±0.04 | 2.1±0.2 | 4.62±0.05 | $(8.6±0.8)×10^{-3}$ | $(6.5±4.0)×10^{-3}$ |
| b2 | -0.78±0.04 | -0.45±0.04 | -2.1±0.2 | -1.69±0.05 | $(-7.9±0.7)×10^{-3}$ | $(-2.2±0.4)×10^{-2}$ |

**4.1 Trends and interannual variability**
The total column $XCO_2$ time series (Fig. 2A) at ALTZ and UNA show a similar mean growth rate around 2.4
ppm/year (2.4 and 2.3 ppm/year for ALTZ and UNA, respectively, Table 3) over the whole measurement period.
A similar mean growth rate is also found for the surface $CO_2$ time series (Table 3 and Fig. 2 B) in ALTZ (2.5
ppm/year). These values are consistent with those estimated at the Mauna Loa Observatory (MLO) reference
station for the 2016-2021 period (average of 2.5±0.5 calculated from surface data available in the NOAA site
https://gml.noaa.gov/ccgg/trends).
At the UNA station a surface mean growth rate of 1.6 ppm/year is found, lower than that observed from the total
column measurements. Comparing the surface mean growth rates with those reported by González del Castillo et
al. (2022) for the 2014-2019 period, we observe a significant difference for the UNA station (2.3 ppm/year in
González del Castillo et al., 2022) but very similar values for the ALTZ station (2.6 ppm/year in González del
Castillo et al., 2022). The difference observed at UNA could stem from (i) starting our new time series at the end



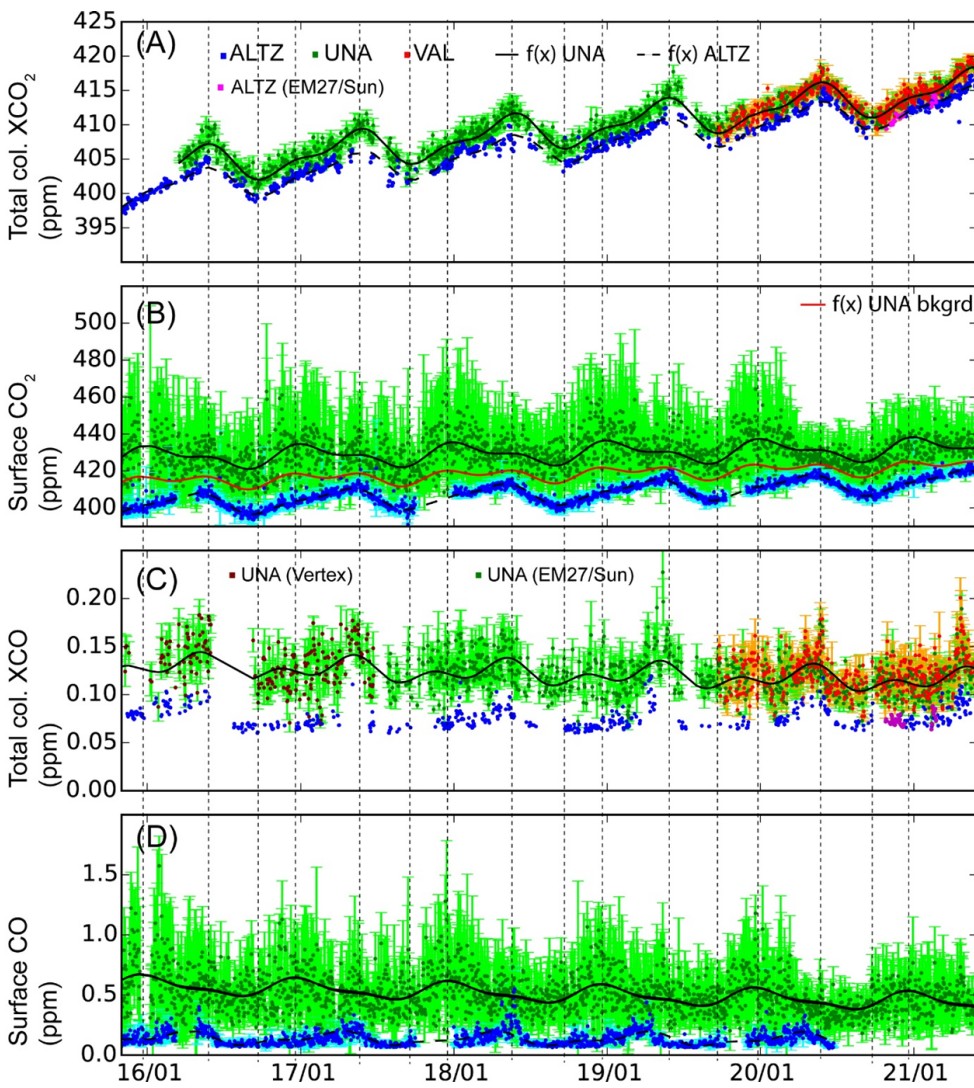


**Figure 2: Time series of (A) the total column  XCO₂ from the FTIR measurements (B) the CO₂ surface concentration from the CRDS measurements, (C) the total column XCO from the FTIR measurements (D) the CO surface concentration from the CRDS measurements for the UNA (in green), VAL (in red) and ALTZ (in blue) stations. For each time series, the daily average data are presented as dots with their daily standard deviations. Black traces show the annual fit calculated from the Fourier series (Eq. (7)). In (A) and (C), we distinguished between ALTZ data obtained from the IFS120/5HR (in blue) and from the EM27/Sun (in magenta) and in (C), between the CO total columns obtained from the VERTEX instrument (in brown) and the EM27/Sun (in green) at the UNA station. In (B) the red curve corresponds to the background fit, calculated following Gonzalez del Castillo et al. (2022), to determine the annual trend and seasonal cycles. Dash lines highlight the minimum and maximum of the annual cycles for the different products.**

of 2015, when the annual growth rate is maximum (González del Castillo et al., 2022) and (ii) the inclusion of the
2019-2021 period, when the mean growth rate clearly decreased. At the VAL station, the total column XCO₂ time
series are found very similar to those observed at UNA stations (Fig. 2A). Figure S1 shows that 86% of the daily
average data at VAL and UNA have a difference lower than 1.0 ppm, although a large part of the comparison was



done during the COVID19 lock-down period (Table1), for which lower gradients are expected due to the decrease
of the anthropogenic emissions.
The interannual variability can be explored through the time series of the mean annual growth rate (AGR) and the
monthly-sampled annual growth rate (MAGR), according to Buchwitz et al. (2018). The MAGR is calculated by
month, as the difference between the monthly-average Xgas data of a year $i$ and the monthly-averaged data of the
previous year ($i$-$1$). The AGR is obtained for each year, averaging all of the MAGR. The AGR and MAGR for
total column and surface measurements are presented in Fig. 4. We include data from the MLO in Fig. 4A, for
which the AGR (dashed black curve) was derived from the surface data available in the NOAA site.
At ALTZ, the interannual variability of the total column $XCO_2$ AGR (Fig. 4A) was found similar to that obtained
from both the ALTZ and MLO surface data, with a coincident peak in 2016, reaching an AGR value of 3.5 (surface
data) and 4.0 (total column data) ppm/year. Surface data AGR time series show a second peak in 2019, which is
not apparent for the total column $XCO_2$ time series. The time series of the MAGR (Fig. 4C) allows better
identifying and characterising the period and duration of the anomalies. The 2016 $XCO_2$ anomaly has a duration
up to 15 months (from October 2015 to March 2017), reaching a maximum value (around 5.0 ppm/year) between
March and July 2016, corresponding to a factor of 2.8 higher than the 2013-2015 base level (1.8 ppm/year).

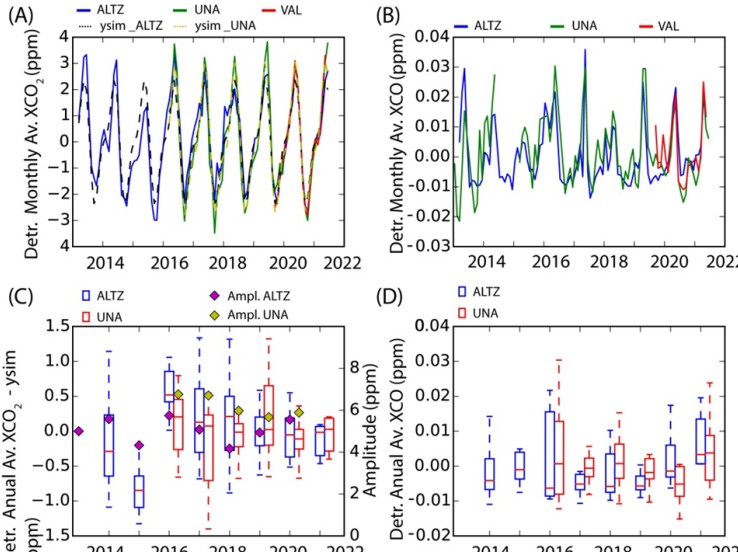

**Figure 3: Interannual and annual variability of the detrended $XCO_2$ and XCO total column data at the UNA, VAL and**
**ALTZ stations. In (C) and (D) the whisker diagrams are calculated from the monthly average detrended data. The**
**amplitude is determined as the max-min values.**

At UNA, the $XCO_2$ AGRs and MAGRs time series (Fig. 4A and C) are very similar to those observed at the ALTZ
station, except for the year 2020. During this year, the AGR dropped by ~20% at UNA before returning in 2021
to the level of the previous two years. This behaviour contrasts with the AGR observed at ALTZ, which remains
nearly constant between 2017 and 2021. The MAGR time series at UNA (Fig. 4C) shows that this drop is
dominated by the exceptionally low June and October growth rates, representing the lowest MAGR values of the





UNA time series. This observation is supported by the VAL MAGR, although the time series is much shorter. The
surface $CO_2$ AGR at UNA shows a much higher interannual variability, with the strongest anomaly observed in
2020, where the AGR is close to zero. A very clear decrease of the day-to-day and intraday $CO_2$ surface variability
is observed in Fig. 2B from April to mid-September 2020, consistent with the $XCO_2$ MAGR anomaly.

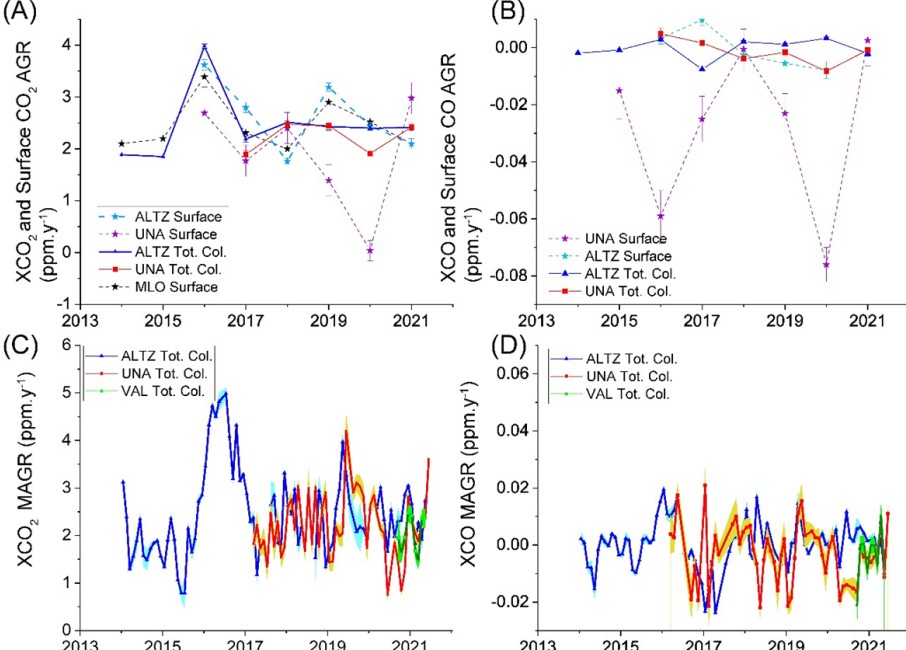


**Figure 4: $XCO_2$ (A) and XCO (B) annual growth rates (AGR) and $XCO_2$ (C) and XCO (D) monthly-sampled annual growth rate (MAGR) obtained from total column and surface measurements for UNA, VAL, and ALTZ stations. In (A), the Mauna Loa (MLO) AGR trend was added in black dash-line. In (A) and (B) errors bars represent the standard error after removing annual cycles, reflecting the data sample quality. The standard error for the MAGR is shown as shaded area in (C) and (D).**


Upon examining CO, the UNA XCO time series (Fig. 2C) has daily averages ranging between 0.10 and
0.23 ppm with a mean and standard deviation of 0.12 and 0.02 ppm, respectively, but shows a decreasing rate (-
$4.0 \times 10^{-3}$ ppm/year) over the whole measurement period. The VAL XCO time series show a very similar baseline
to UNA, with a daily average difference lower than 0.02 ppm for 85% of the coincident dataset (Fig. S1). At the
ALTZ background site, the XCO baseline and day-to-day variability are lower than at UNA and VAL, as expected
(mean and standard deviation equal to 0.08 and 0.01 ppm, respectively). The surface CO time series (Fig. 2D)
shows a more significant decreasing trend ($-2.68 \times 10^{-2}$ ppm/ year) than the total column data at UNA, while the
baseline at ALTZ remains constant around 0.11 ppm. The CO AGR and MAGR at ALTZ and UNA are shown in
Fig. 4B and D. Generally, the XCO AGR and MAGR oscillate around their base level at the ALTZ and UNA
stations, with short-term anomalies. At ALTZ, a strong negative XCO AGR anomaly is observed in 2017, which
was not observed for $XCO_2$, likely resulting from the exceptionally high XCO columns measured during 2016.
This is supported by the increase of the XCO MAGR from October 2015 to July 2016 (Fig. 4D), coinciding with
the first 10 months of the highest $XCO_2$ anomaly and followed by the lowest XCO MAGR values of the time series



(around -0.02 ppm/year in April 2017). At the UNA station, the AGR slightly decreases between 2016 and 2020
and increases again in 2021. The most significant and prolonged (>5 months) MAGR anomaly (Fig. 4D) occurred
between April and September 2020, with negative values. Some short-term additional anomalies are observed, but
only a few of them (in May 2018 and January 2019) are not affected by the limited number of available
measurements.
**4.2 Seasonal variability and short-term cyclic events**
Annual cycles are observed for both total column $XCO_2$ and $CO_2$ surface measurements at ALTZ, UNA and VAL
stations (Fig. 2). The minimum and maximum of the total column $XCO_2$ cycles are observed in May-June and
September, respectively, with an average amplitude around 5 (ALTZ) and 6 (UNA) ppm.
To examine the temporal changes in amplitude and shape of the annual cycles, total column data were monthly-
averaged, detrended by subtracting the linear part of the fit ($f(x) = ax$, in Eq. (7)), and compared to the detrended
mean annual cycle ($f(x) - ax$) in Fig. 3. To obtain a longer-term view, we included the 2013-2015 period from the
ALTZ station, previously published in Baylon et al. (2017), after applying the inter-calibration factors (section
3.1.3). At ALTZ, two periods significantly deviated from the average $XCO_2$ seasonal cycle, i.e.: (i) the year 2015,
where all the monthly averaged $XCO_2$ are below the fit and with one of the lowest seasonal amplitudes (~4.0 ppm)
of the whole time series, and (ii) the year 2016, with higher monthly averages than the mean $XCO_2$ seasonal cycle
and the highest amplitude (~5.8 ppm). At UNA, the difference with respect to the average $XCO_2$ seasonal cycle is
not significant, except for the year 2020, where all the monthly averages are below the mean annual cycle. During
this period, the UNA and VAL $XCO_2$ monthly-averaged data fit exceptionally well with those of the ALTZ station
between March 2020 and March 2021 in terms of shape and amplitude, while the UNA and VAL annual cycle
amplitudes are slightly higher than those of ALTZ for the other years.
Regarding the $CO_2$ surface data (Fig. 2B), annual cycles are observed with maxima and minima reached mid-
December and mid-September, respectively. As also reported in González del Castillo et al. (2022), the maximum
occurred during winter, when shallower boundary layer prevails and the summer-autumn minimum can be
explained by the dilution of trace gases in a deeper convective boundary layer and more active urban vegetation.
XCO peaks every year in April-May at the three stations (Fig. 2C and Fig. 3B) and then shows minimal annual
values in August, preceding by 1 month the minimum and maximum values of the $XCO_2$ time series. The April-
May maximal annual values, also confirmed by TROPOMI measurements (Borsdorff et al., 2020), coincide with
the biomass burning season and the periods during which the mixed layer reaches its maximum altitude (García-
Franco et al., 2018). During 2015, the XCO time series show a very low maximum reached in February instead of
May (Fig. 3B), contrasting with 2016, where high total column XCO values are reached in January and maintained
for a period of at least 5 months. Additionally, in 2018, the XCO annual cycles differ from the other years with
lower values and a flat shape during the first semester of the year (January-May).
Surface CO data (Fig. 2D) also show periodic increases at the ALTZ station with maxima reached during April-
May, coinciding with the maxima observed from total column XCO measurements. They confirm the increase of
the CO emissions during the biomass burning season, at least dominant in the ALTZ measurements. However, at
the UNA station, cycles are also observed in the surface data but with a maximum coinciding with that of the $CO_2$
surface data, and lagging behind the XCO total columns. These cycles are likely dominated by other processes
affecting both CO and $CO_2$ species such as the mixed layer seasonal dynamic.



### 4.3 Intraday variability

The intraday variability of the total columns and surface data are depicted in Fig. 5 and Fig. 6. Since the ALTZ
total column data do not present a significant diurnal pattern (the hourly variability remains lower than the standard
error of the time series), they are not presented in these plots.

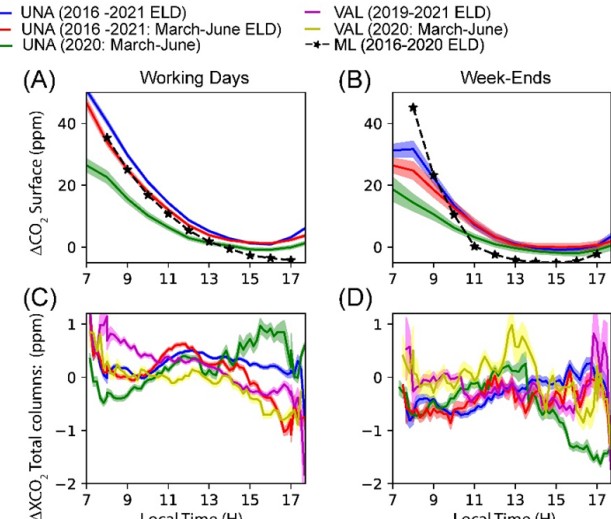


**Figure 5: Diurnal patterns of the detrended surface $CO_2$ mole fractions (A and B) and $XCO_2$ total columns (C and D) measured at UNA and VAL stations. For each panel, the different curves represent different time periods: in blue, the whole measurement period excluding the lock-down period (March-June 2020 ELD), in green the lock-down period (March-June 2020) and in red the whole measurement periods from March to June, excluding the lock-down. The standard errors are presented as shaded areas. Black curves represent the diurnal pattern of $CO_2$ in the Mixed Layer (ML) calculated from the total columns data for the UNA station.**

Total column data were detrended by removing the seasonal fit (black traces in Fig. 2A and Fig. 2C), and averaged
over 10 min. To avoid a possible bias due to strong ventilation periods, a filter based on a ventilation index (VI)
was applied, following recommendations in Hardy (2001), Su et al. (2018) and Storey and Price (2022). The VI is
calculated as the product of average wind speed velocity (between the surface and 100 m height), and the planetary
boundary layer height for UNA and VAL locations. The wind velocity and the MLH were estimated with the U
and V wind components and the PBL height fields from the hourly ERA5 reanalysis product (Hersbach et al.,
2020). In the MCMA, the surface wind speed presents a diurnal pattern, generally reaching a maximum during the
afternoon between 14 and 15h LT (Fig. S4). The filter selects the days complying with the following criteria (i) a
maximum wind velocity (average 10-100m height) between 10h and 12h LT lower than 1.5 m.s$^{-1}$ (threshold based
on Stremme et al., 2013) and (ii) a daily VI lower than 2350 m$^2$.s$^{-1}$, which represents a commonly used threshold
for selecting poor ventilation conditions (Hardy, 2001; Storey and Price, 2022). About 60% of the original $XCO_2$
and XCO dataset is selected by applying the filter, and will be considered in the following analysis. We note that
about 70% of discarded data corresponds to the January-May period of the year. Filtered total column $XCO_2$ and
XCO data were averaged by 10 min and presented in Fig. 5C-D and Fig. 6C-D, distinguishing between the working
days (WD) and the week-end (WE) periods. To explore the 2020 lock-down influence on the diurnal pattern, three
different periods were distinguished for each plot, the first one (blue trace) corresponding to the whole
measurement period excluding the interval in 2020 during which a significant MGRA decrease was observed





(March - October 2020 ELD) due to the lock-down period; the second (green trace: March- June 2020) only
includes the lock-down period, and additionally excludes the rainy season to avoid bias due to incomplete daily
time series; and the third period (red trace) is the same as the first one, but only considering the March to June
months to be compared with the lock-down period.

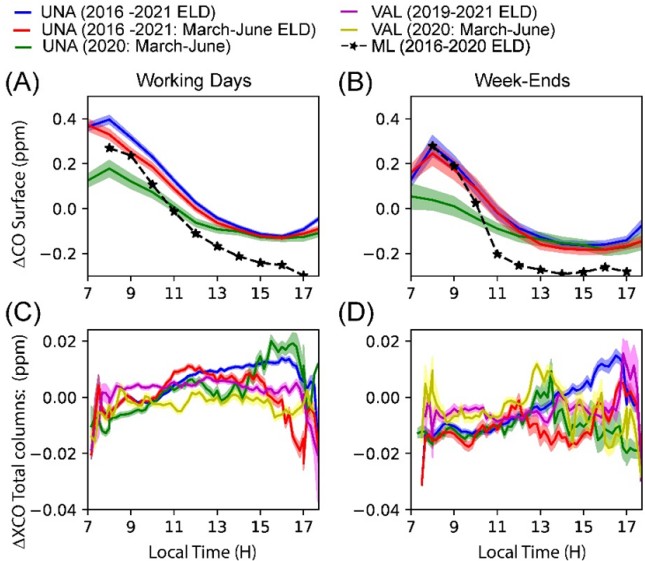


**Figure 6: Same as Figure 5, but from surface CO and total column XCO measurements.**
Surface data from the CRDS analyzers were detrended by removing the background fit following the methodology
described in the section 3.2, and filtered to be coincident with the filtered total column measurements (selection of
data between 7 and 18 h LT and only including the days with low ventilation conditions). They were finally
averaged by hours and presented in Fig. 5A-B and Fig. 6A-B for the WD and WE periods, respectively, for which
each curve represents the periods mentioned above.

The surface $CO_2$ diurnal pattern at UNA station for the whole measurement period (2016-2021, Fig. 5A and B in
blue) is consistent with the one previously described in Gonzalez del Castillo et al. (2022) for the 2014-2019
period, with a maximum observed during the early morning (reached before 7 h LT), a minimum during the
afternoon (between 15 and 16 h LT) and an average amplitude around 45 ppm. A lower amplitude of these cycles
is observed at WE (average amplitude of 28 ppm) with respect to the WD periods. During the 2020 lock-down
period (green curve), the WD surface $CO_2$ diurnal profile has a comparable amplitude (average amplitude of 26
ppm) to those of the WE for the whole measurement period, and slightly higher than that observed during the lock-
down WE periods (average amplitude of 22 ppm). The surface CO diurnal profile (Fig. 6: 2016-2021, blue curve)
peaks at 8h and then decreases until 16 h LT during any day of the week. The WD and WE data shows amplitudes
of up to 0.5 ppm and 0.3 ppm, respectively. During the lock-down period the WD and WE amplitudes are much
lower (0.3 and 0.2, respectively), consistently with the $CO_2$ surface observations.
The total columns $XCO_2$ and XCO diurnal patterns (Fig. 5C-D and Fig. 6C-D) have very different shapes than
those of the surface data, with amplitudes one order of magnitude lower. The variability observed between 7 and



8h is likely due to the low number of measurements during this time interval, and will not be taken into account in
the following analysis. The UNA and VAL $XCO_2$ diurnal patterns significantly differ in shape. The VAL WD
curve (magenta trace) continuously decreases from 8h to 17h (amplitude around 2 ppm) during both the whole
measurement and lock-down periods, but during the lockdown period, lower values are generally recorded with
higher intra-hour variability between 11h and 14h. The general WD decreasing trend suggests a maximum reached
during the early morning (before 7 h LT). This observation is supported by the $CO_2$ surface measurements
performed with the low-cost medium precision $CO_2$ sensors (Grutter et al., 2023), recording a maximum between
6h and 7h LT. The UNA $XCO$ WD diurnal pattern (blue trace) is almost constant until 10h, then increases until
reaching a maximum around 12h, slightly decreases until 17h LT and finally shows an abrupt decrease after that.
The amplitude of the diurnal variability is around 1 ppm. During the lock-down period, the diurnal profile is
different, increasing until 12h LT, slightly decreasing until 13h LT and then increasing again until reaching a
maximum at 16h, and finally abruptly decreasing until 17h LT. The lock-down WD $XCO_2$ profile shows lower
values than the other periods until 13h, but the peak observed at 16h is not apparent for the other periods.
Variability is generally lower at WE (<1ppm), except for the lock-down period, for which an important decrease
is observed after 14h LT, but it is likely affected by a low number of measurement days. For XCO, the diurnal
profiles also have different shapes at UNA and VAL. At UNA, the March-June XCO diurnal profiles (red and
green curves) resemble that of $XCO_2$ for both the lock down and whole measurement periods. When considering
the twelve months of the year (blue trace), the maximum curve slightly increases between 12h and 16h, when it
reaches its maximum. It contrasts with the variability of the March to June months curves during this time interval,
for which an increase is observed during the lock-down period or a decrease if considering the whole measurement
period. At VAL, the diurnal profile is fairly constant until 17h with slightly lower values during the lock-down
period.
The total column $XCO$ diurnal profiles at WE are less reliable with larger standard errors, likely due to the low
number of considered measurements. An increase is nevertheless observed at UNA where the considered day's
number is statistically more reliable, with a peak around 17h LT, which was not observed for $XCO_2$.

### 580    4.4 CO and $CO_2$ within the mixed layer from FTIR and surface data.

Figure 7 shows the hourly-averaged $CO_2$ and CO concentration within the mixed layer ($CO_2^{ML}$ and $CO^{ML}$
products), calculated from the FTIR measurements (see section 3.4), concurrently to the surface data. The $CO_2^{ML}$
and $CO^{ML}$ products are in agreement with the surface observation, with a slope of 0.95±0.02 ($R^2$=0.74) for $CO_2$
(Fig. 7C) and 0.81±0.02 ($R^2$=0.74) for XCO (Fig. 7D). For $CO_2$, the slope was found closer to 1.0 (1.00±0.02)
with an offset of -2.9±0.2 and a better $R^2$ (0.77) when discarding the data corresponding to the rainy season. This
effect is likely due to the removal of the incomplete daily time series frequently interrupted at the beginning of the
afternoon during the rainy season.



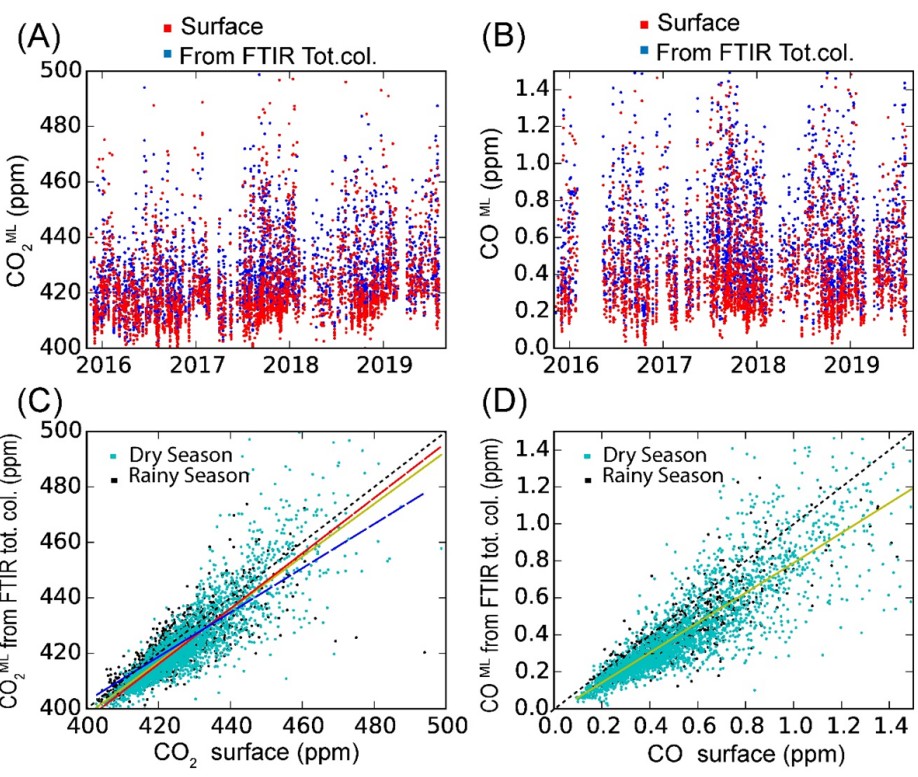

**Figure 7: Comparison between (A) the CO$_2^{ML}$ and (B) CO$^{ML}$ products (red) derived from the ALTZ and UNA total column measurements and the surface measurements at the UNA station (blue). (C) and (D) represent the correlation plots for CO$_2$ and CO, respectively. In (C) and (D), we distinguished between data corresponding to the dry (November to May: cyan) and rainy (June to October: black) seasons. In (C), yellow, red and blue linear regression curves correspond to the whole measurement period (yellow: slope=0.95±0.02; Offset= 17.9±0.2; R$^2$=0.74), the dry season (red: slope=1.00±0.02; Offset: -2.9±0.2; R$^2$=0.77) and the rainy season (blue: slope=0.80±0.03; Offset: 83.7±0.39; R$^2$=0.66). In (D), since no significant difference was found for the different period, the regression line (yellow: slope=0.81±0.02; offset: -0.021±0.004; R$^2$=0.74) represent the whole measurement. The black dash line represents y=x.**

The CO$_2^{ML}$ and CO$^{ML}$ diurnal patterns are presented in Fig. 5 and Fig. 6 (dash lines) together with those of surface measurements, after a similar filtering. The CO$_2^{ML}$ and surface CO$_2$ diurnal patterns (Fig. 5A) are very similar in shape and amplitude, especially during the WD, although a small difference is observed at the end of the afternoon (<5 ppm). This difference is likely due to the increase of the uncertainties of the MLH estimate when it is more diluted. The CO$^{ML}$ and surface CO diurnal profiles (Fig. 6A) also have similar amplitudes and shape for both WD and WE, although the CO$^{ML}$ diurnal profile shows lower values (offset around 0.1 ppm at WD). Despite this very simplified model, these results show that the total column and surface measurements are mutually very consistent when the seasonal and diurnal variability of the ML expansion above Mexico City is taken into account.



**4.5 XCO₂ to XCO enhancements ratios**


The XCO and XCO₂ correlated enhancements and their ratio can give insights into the combustion efficiency of
the sources in a city, and therefore on their contributions. In this study we explored the variability of the
XCO/XCO₂ ratios at both long-term and intraday scales.
For the long-term analysis, the XCO₂ "background" level was calculated using a statistical method, using the lower
5$^{th}$ percentile of the measured Xgas over a 1-day running window (You et al., 2021). We did not use the ALTZ
measurements because of (i) the periodic influence of the wildfires in the region during the dry season, and (ii) the
discontinuity of our daily averaged time series. The enhancements above background $\Delta_m XCO_2$ and $\Delta_m XCO$
measured at UNA and averaged by months and their ratios are presented in Fig. 8, as whisker diagrams.

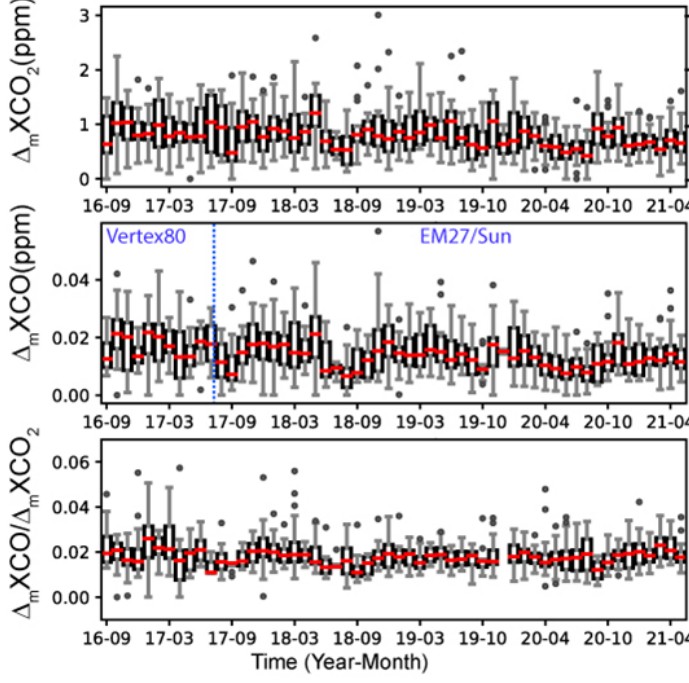


**Figure 8: Whisker diagram representing by month the variability of ΔXCO₂, ΔXCO and their ratio from the UNA**
**measurements.**
Both $\Delta_m XCO_2$ and $\Delta_m XCO$ time series show a slight decrease over time (around 0.05 ppm/year and 0.001
ppm/year, respectively). Although the $\Delta_m XCO/\Delta_m XCO_2$ ratio displays a variability around its mean value
(0.018±0.003), there are no discernible cyclic or long-term trend in the time series, except for the rainy periods of
2017, 2018 and 2020 when low ratios (and low $\Delta_m XCO$ and $\Delta_m XCO_2$ values) were observed. The $\Delta_m XCO$ and
$\Delta_m XCO/\Delta_m XCO_2$ ratio show a higher variability at the beginning of the time series (until July 2017) likely due to
the use of the CO Vertex products.

To perform the intraday analysis, the hourly-averaged data were first detrended by subtracting the daily average.
The resulting $\Delta_i XCO_2$ vs. $\Delta_i XCO$ datasets are plotted in Fig. 9A. The entire $\Delta_i XCO_2$ and $\Delta_i XCO$ datasets showed
a good correlation at both the UNA and VAL stations, with similar linear regression slopes around 0.0164±0.0003,



which is consistent with that found from the surface measurements and the ML product (Fig. 9B). Although there
is an actual difference in the emission types of the southern and northern parts of the city, the North hosting
industrial and commercial sources and the South being largely residential and commercial, the common and
dominant source of CO in the MCMA (at UNA and VAL stations) seems to incriminate motorised vehicles. The
data dispersion around the regression line likely reflects more punctual and local influence of other sources with
an important week-to-week variability.

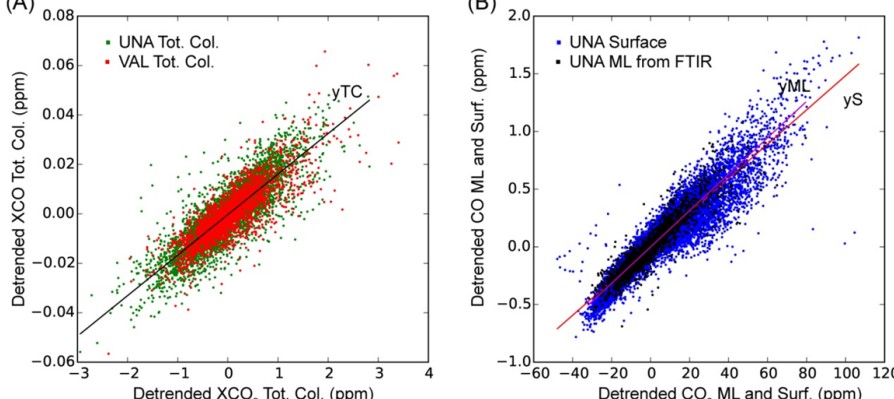



**Figure 9: A: Correlation plot of (A) the detrended (by removing the daily averages) hourly-average total column $XCO_2$ vs. XCO data, and (B) the detrended hourly average Mixing Layer (ML) and surface $CO_2$ vs. CO products. Solid lines represent the linear regression lines, with the following parameters: TC slope=0.0164±0.0003, $R^2$=0.72 for the total columns at UNA and VAL; yS slope=0.0148±0.0001, $R^2$=0.87 for the surface products and yML slope=0.0158±0.0002, $R^2$=0.88 for the Mixing Layer products.**

On the other hand, the total column (UNA-VAL) differences, presented in Fig. S3 can also be used to calculate
the $\Delta XCO/\Delta XCO_2$ ratio, with a more precise subtraction of a common background (which assumes a
homogeneous background across the entire city) from the two stations. Figure 10 shows the hourly-average $\Delta XCO_2$
(UNA-VAL) vs. $\Delta XCO$ (UNA-VAL) correlation plot for the coincident measurement period. A well-defined
linear correlation is observed with a slope of 0.015±0.001 and a coefficient of determination of $R^2$=0.80, highly
consistent with that found in Fig. 7. The use of the (UNA-VAL) total columns difference notably improved the
coefficient of determination, by removing the regional long-term and short-term perturbations affecting the two
sites. The intraday variability of the $\Delta XCO$ (UNA-VAL)/$\Delta XCO_2$ (UNA-VAL) ratio (Fig. 10: colour scale),
showing higher columns at VAL during the morning and at UNA during the afternoon likely reflect the South to
North transport of air across the city. We note that the ratio remains the same during the lock-down period. We
would expect lower intraday (UNA-VAL) $\Delta XCO$ and $\Delta XCO_2$ amplitudes during the lock-down period, but it is
not clearly apparent in this correlation plot.




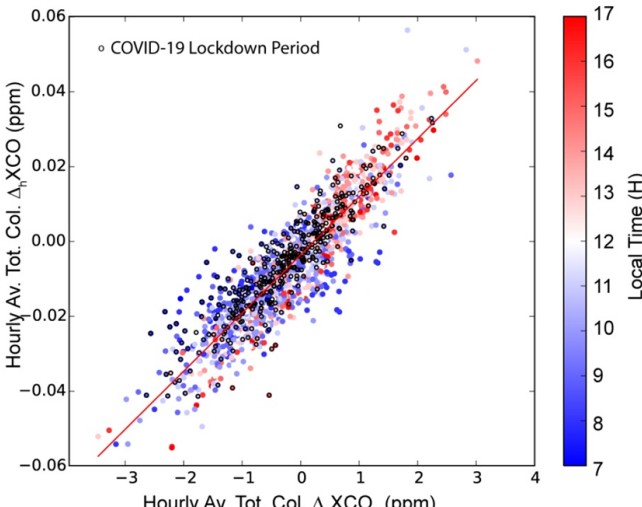

**Figure 10: Correlation plot of the ΔXCO (UNA - VAL) vs. ΔXCO₂ (UNA - VAL) hourly averages (colour scale depending on the time is shown to the right) for the coincident measurement period (September 2019 - June 2021). Dots with black edges highlight the measurements during the COVID19 lock-down period (March-June 2020). Regression line (in red): Slope: 0.015±0.001, R²=0.80.**

**4.6 Estimate of CO and CO₂ MCMA emissions.**

The variability of the long-term CO emissions in the MCMA can be estimated, following the method detailed in Stremme et al. (2013). In that study, they assumed that, since the XCO emissions in the MCMA are mainly due to traffic pollution, the rapid changes observed in the XCO total column (less affected by the airmass vertical distribution) should reflect the CO fresh emissions under certain meteorological conditions. Low ventilation, strong turbulence in the mixed layer and limited zenithal angle of measurements are critical criteria to avoid enhancement due to horizontal transport or local heterogeneity. XCO growth rates can be estimated at specific time intervals complying with these conditions from long-term time series. Further details on the method and estimates of uncertainties due to these assumptions are given in Stremme et al. (2013). Here, we determined an optimised time interval for estimating the mean CO growth rate using (i) the diurnal surface wind speed patterns and (ii) the MLH growth rate, the latter reflecting the turbulence within the mixed layer (Fig. S4). The time interval complying with a rapid growth of the mixed layer and low surface wind speed (< 2 m.s⁻¹) was found between 10 and 12h, which is in agreement with the requirements mentioned in Stremme et al. (2013). Growth rates and their uncertainties were determined by year, based on the linear regression (with 95% confidence interval) of the 10-min averaged detrended CO total columns over the 10-12h interval. For example, for the year 2018, we found a CO growth rate of 52±5 kg.km⁻².h⁻¹.

To extrapolate the growth rate over the MCMA, we used the TROPOMI CO total column data that we averaged over the 2018-2022 period (Fig. 1), following the same method as described in Stremme et al. (2013). We assume that the total amount of fresh CO is proportional to the total emission of the MCMA and to the total column enhancement at the UNA site, which reflects the CO accumulated at this site. The ratio of the total accumulated CO in the MCMA to the accumulated CO at UNA is therefore the same as the emission ratio of the whole Megacity





to the emission flux at UNA. Therefore this ratio is the extrapolation factor and represents an effective area, defined
as Eq. (8):
$$Eff\_Area = \frac{\int (CO_{MCMA} - CO_{bgrd})}{CO_{UNA} - CO_{bgrd}} \qquad (8)$$

As the TROPOMI overflight time is around 13h30 LT, we cannot neglect the ventilation and slight advection is
smoothing out the distribution, so that both the background and the column at UNA have to be chosen carefully.
The background column was therefore estimated in two ways (i) from the smallest value observed upwind the city
at the elevation of the Mexican basin (contour line separating Mexico City from the Toluca area in the west in Fig.
1) and found to be $1.45 \times 10^{18}$ molec.cm$^{-2}$ and (ii) from the Tecamac site, where the border of MCMA was assumed
in the Stremme et al. (2013) and where the column was found to be $1.60 \times 10^{18}$ molec.cm$^{-2}$.
Due to advection, even locations slightly out of the megacity are presenting enhanced CO columns and it is not
clear which is the background column in the Mexican basin. Figure S5 illustrated the sensitivity of the effective
area to the background uncertainties. A 10% higher background leads to a 40% smaller extrapolation factor and a
40 % emission underestimate.
The mixed layer column at UNA from the TROPOMI data was found to be $1.93 \times 10^{18}$ molec.cm$^{-2}$ (Fig. 1), which
is consistent with our EM27/SUN ground-based measurements (average of $2.17 \times 10^{18}$ molec.cm$^{-2}$). In cases where
the CO total column is lower than the background, likely due to the topography effect, we set the difference column
to zero for the integration. This topographic effect is important for the considered area, as there are plenty of
mountain around the basin, like the mountain ridge in the west (including Ajusco, Desierto de Leones, etc.), some
mountains in the mountain ridge on the eastern part of the area including in the south the two volcanoes
Popocatépetl and Iztaccihuatl. Finally, we found effective areas of ~2017 km$^2$ (outer area) and ~1178 km$^2$ (inner
area) considering the two background values given above. The "inner area" reflects conditions without ventilation
effect, therefore the outer area is more appropriate for the emission estimates given that the TROPOMI
measurements occurred at 13:30 when the ventilation cannot be neglected. The other estimates calculated from the
inner area will be thereafter only indicated within brackets and considered to estimate the sensitivity of the result.
Since the measured growth rate corresponds to a time interval of only 2 hours in the middle of the day, the CO
intraday fluctuations have to be taken into account. Stremme et al. (2013) used a factor which was taken from the
available bottom-up inventories and described that the CO emissions per/day are roughly 18.5 times the emission
per hour at noon. Assuming the same factor, we estimate a CO rate around 0.71±0.06 (0.42±0.04) Tg/year for
2018. If no information about the diurnal distribution of the emission rate would be available, we should assume
a uniform distribution and an upper value of the CO rate could be estimated using an intraday time interpolation
factor of 24 hours instead of 18.5, finally resulting in ~30% higher estimates. Despite the significant uncertainties
introduced by spatial and temporal interpolation, their impact on the relative variability, trends and anomalies of
the emission rates is less important if the same method and assumptions are consistently applied across the entire
time series.
$CO_2$ emissions could not be directly estimated using the same method, given its complex diurnal pattern, which is
a cumulative result of both natural and anthropogenic contributions and likely been influenced by additional
factors, related to instrumental and retrieval effects. Instead, we based our $CO_2$ estimates on the measured
XCO/XCO$_2$ ratio. The average XCO/XCO$_2$ molec. ratio (0.0164±0.0003) determined from the UNA and VAL



total column measurement (Fig. 9) was converted to a mass ratio (multiplying it by the molecular weight ratio)
and found to be 0.0100±0.0002. Considering this ratio, we estimated the $CO_2$ annual emission at 71±6 (42±4)
Tg/year for 2018. Our estimates of CO and $CO_2$ emissions by year and their average over the whole time series,
applying the same method, are presented in Fig. 11 and Table S3, concurrently with the SEDEMA inventories for
the MCMA. We obtained a 2016-2021 CO and $CO_2$ average emissions of 0.55±0.02 (0.32±0.01) and 46±2 (32±1)
Tg/year, respectively, when excluding the lockdown period (Table S3). Here, the given uncertainties are solely
those stemming from the propagation of errors in growth rate estimates. Uncertainties on absolute values are much
higher when considering spatial and temporal extrapolations errors, but they do not influence the interpretation of
relative values.

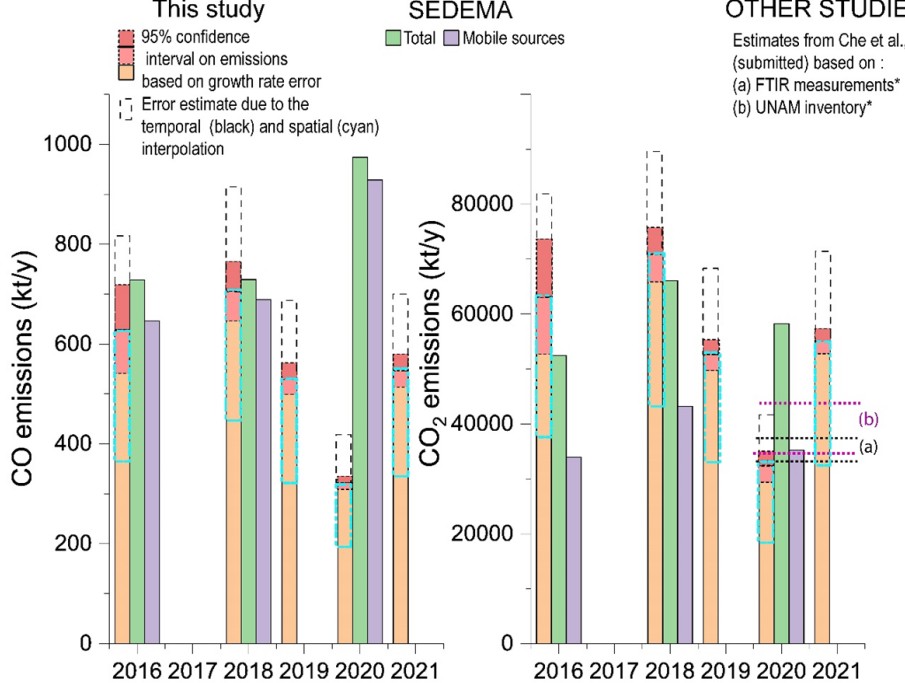

*: The same intraday temporal interpolation factor was applied for the comparison. (a) and (b) were based on the 10/2020 - 05/2021 period
***Figure 11: Comparison of CO and $CO_2$ emission estimations from UNA FTIR diurnal growth rates and from SEDEMA***
***inventories. For $CO_2$ (right), the estimates from Che et al. (submitted) are also reported, although it was based on the 10/2020***
***to 05/2021 period, after applying the same intraday temporal factor as used for our study to convert the Gg/hour to t/year.***
**5 Discussion**
**5.1 Long term variability**
In this contribution, we characterised the seasonal and inter-annual variability and trends of the CO and $CO_2$ total
column and surface concentrations from two urban and one background stations. The average total column 2013-
2019 growth rate obtained at ALTZ (~2.5 ppm/year) and its inter-annual variability are in accordance with that
typical of the Northern Hemisphere measurements from TCCON stations (hereafter, NH-TCCON) (Sussman et
al., 2020: AGR of 2.4 ppm/year for the 2012-2019 period).





Both the NH-TCCON and ALTZ stations captured an important increase of the AGR in 2016 (+1.1 ppm/year for
the TCCON stations and +2.1 ppm/year for the ALTZ station with respect to 2015), coinciding with the most
intense ENSO (El Niño Southern Oscillation) event since the 1950s'. The impact of "El Niño" events on the carbon
cycle is not yet fully understood, although they are consistently accompanied by a global increase of $XCO_2$ due to
increasing drought in many regions and a decrease in global land carbon uptake. In 2016, an increase of 1.3
ppm/year was observed in the Mauna Loa in situ AGR with respect to 2015 (Betts et al., 2018), for which the
contribution of the 'El Niño' event was estimated at about 25%, the rest ascribed to an increase of the
anthropogenic emissions. In Mexico, the "El Niño" events are generally associated with a decrease in
precipitations, with deficits which can reach up to 250 mm in the South-Western area of the country, causing
drought and a higher occurrence of wild and forest fires (Bravo-Cabrera et al., 2018, González del Castillo et al.,
2020). Our observations from the ALTZ measurements highlight a much higher $XCO_2$ increase (+2.1 ppm/year)
during 2016 with respect to 2015 than that observed at the NH-TCCON stations. During this period a small increase
in the XCO MAGR (~ +0.02 ppm) is also observed at both ALTZ and UNA stations, maintaining the highest
values of the whole time series over more than 4 months. Assuming that the CO MAGR variability captured at the
ALTZ station during 2016 rather reflects a change in the global MCMA's emissions, we attempt to delineate the
global and local contributions in the 2016 $XCO_2$ ALTZ AGR increase. Adopting a molecular $CO/CO_2$ ratio of ~
0.016, a hypothetical increase of the $XCO_2$ MAGR over the 09/2015 - 09/2016 period due to the local emissions
would be around +1.2 ppm/year, thus about 60% of the observed increasing rate during this period (+2.1 ppm/year).
This gross estimate suggests that the El Niño regional effect only contributed at about 25% (0.9 ppm) to the
observed GRA increase, which is close to the estimate from the NH-TCCON stations (~ +1.1 ppm) and from in
situ data.
On the other hand, our long-term FTIR and surface time series allows examining the effect of the COVID-19 lock-
down on the tropospheric $CO_2$ and CO concentration above the MCMA at local and regional scales. The reduction
of the surface CO and $CO_2$ AGR at UNA ($CO_2$ AGR to a value close to zero, and CO AGR ~ -0.1 ppm/year) with
respect to the other years (Fig. 4), and the strong diminution of their amplitude in the mean diurnal cycles clearly
reflect a significant decrease of the local emissions near the UNA station, likely due to a drastic reduction of the
urban traffic (the average annual congestion level decreased from 52% in 2019 to 36% in 2020 in Mexico City,
from TomTom available estimates https://www.tomtom.com/traffic-index/mexico-city-traffic/).
The FTIR total column $XCO_2$ and XCO time series at UNA did not capture such a drastic change, only a small
punctual decrease of the MAGR lower than the standard deviation of the whole time series was observed between
April and October 2020. These results are in accordance with previous studies in other parts of the world. Although
a reduction of 8.8% of the global $CO_2$ emissions was observed during the first five months of 2020 (Liu et al.,
2020; Jones et al., 2020) and an annual reduction from 4 to 7% (Le Quéré et al., 2020), the atmospheric total
column $XCO_2$ showed a less clear effect (Sussman et al., 2020).
**5.2 $CO/CO_2$ ratio and MCMA emission estimates**
In this study, we robustly determined the $CO/CO_2$ ratio characterising the combustion efficiency of the city
(0.016±0.01) from both surface and total column measurements at two urban stations. We found the same ratio for
the UNA and VAL stations, and this ratio is very consistent with that found using the (UNA-VAL) gradients and
using the surface measurements. This ratio is also consistent with that reported by MacDonald et al. (2023),



calculated from TROPOMI and OCO-2/3 measurements (0.019) and slightly higher than that obtained from the
EDGAR, FFDAS and ODIAC inventories (ratio ~0.012) reported in the same study.

780        Our estimate of CO emissions from the UNA measurements is based on a simplified approach, limited to

days with low ventilation and time intervals corresponding to the late morning hours. It assumes a homogeneous
area in the footprint and averages selected days without discrimination. Given that the temporal and spatial
extrapolation introduces large uncertainties, only the relative and interannual behaviour of the emission can be
discussed here, but the approach demonstrates how close column growth rate can be related to emission flux, if
meteorological conditions allow neglecting advection. Our estimated range of CO emissions are consistent with
the SEDEMA inventories at least for the year 2016 (factor 0.98) and 2018 (factor 1.04) if considering that they are
dominated by the mobile sources. However, it is not the case for 2020, for which our estimate is much lower than
SEDEMA by a factor of 0.3. During the lock-down period we estimated a decrease of about 55% compared to
2018 while in the SEDEMA report, 2020 is the year with the maximum CO emissions (increase of 35% compared
to 2018 considering the mobile sources). Both of these estimates contrast with Kutralam-Muniasamy et al. (2021),
which reported an increase of 1.1% during the lock-down using the RAMA surface data. The large difference
observed between these different studies can be due to i) the different methods used for extrapolating in space and
time the emissions, ii) higher uncertainties of the FTIR-based estimates due to an important reduction the selected
days of measurements and iii) an over-estimation of the SEDEMA inventory due to a lack of data during the lock-
down period. Our estimate is based on the extrapolation of data from only one station (UNA), for which the
dominant source is mainly the UNAM traffic activity. During the lockdown, the UNAM was closed and a
significant reduction of the local traffic was recorded, but this traffic reduction was likely not representative of the
whole MCMA. However, the decrease of the MGRA at both VAL and UNA stations does not support the increase
of the CO emissions estimated by the SEDEMA inventory. Interestingly, it was not possible to apply the same
method to calculate CO emissions at VAL because the average growth rate was close to zero (Fig. 6). This
behaviour at VAL is likely due to the fast dispersion of the pollutant at this site, weakening the link between the
diurnal pattern and the emissions.
Regarding $CO_2$, our estimates also agree with the SEDEMA's inventory, especially if we consider the total
emissions instead of mobile sources (factor of 1.2 and 1.1) for the years 2016 and 2018. For 2020, we estimated a
decrease of 55% while the SEDEMA inventory indicates a decrease of about 10%. The SEDEMA $CO/CO_2$ ratios
for total emissions are similar to ours (0.014 and 0.011 in 2016 and 2018, respectively). This result suggests that
the $CO/CO_2$ ratio we determined does not exclusively reflect the traffic emissions, as expected, but should have
another component. The SEDEMA ratios are higher for mobile sources (0.019 and 0.016 in 2016 and 2018,
respectively) and one order of magnitude lower for industrial and domestic burning, which represent the second
main $CO_2$ anthropogenic sources. If we consider the 2018 SEDEMA ratio for mobile sources (0.016), we find $CO_2$
emissions of the order of 43,100 kt/year for this year, within ~5% of the SEDEMA estimates.
Our results were also compared with the estimates reported in Che et al. (submitted), based on an intensive FTIR
measurement campaign performed during the 10/2020 to 05/2021 period and using a Column-Stochastic Time-
Inverted Lagrangian Transport model (X-STILT) and a bayesian inversion (Fig. 11). Considering the same
measurement period, our method leads to $CO_2$ emission estimates ranging between 29,000 and 49,800 kt/year
using inner and outer effective area, respectively, which is consistent with the estimates obtained in Che et al.
(submitted), ranging between 32,700 and 37,200 kt/year when applying the same intraday temporal extrapolation



factor. Although the method we used for estimating the MCMA emissions is coarse and contains large
uncertainties, mainly due to the temporal and space extrapolation, it shows the ability to use one station capturing
the variability of the anthropogenic emissions of the MCMA and providing a year-by-year follow-up emission
information without using complex dispersion models.

## 6 Summary and conclusion

We have analysed the variability of the total column XCO and $XCO_2$ above the MCMA from two urban and one
background stations. The long-term $XCO_2$ data at the ALTZ station shows an average annual growth rate of ~2.5
ppm/year, similar to what has been reported from TCCON stations in the northern hemisphere, and captured the
perturbation driven by the 2015-2016 El Niño event. The urban stations show a similar growth rate (~2.3 ppm/year)
and unlike at ALTZ, a slight decrease of $XCO_2$ and XCO during the COVID19 lock-down period could be
observed. The $CO_2$ and CO concentrations within the mixed layer, estimated from the FTIR total column
measurements and ceilometer data, were found to be consistent with the surface measurements. These findings
confirm that the concentrations near the surface are mainly controlled by the emissions and the daily behaviour of
the mixed layer in MCMA. Our long-term total column and surface time series from both urban stations allowed
us to determine with great confidence an average $CO/CO_2$ ratio, indicative of the Mexico City combustion
efficiency. The $CO/CO_2$ ratio over our long-term measurement period seems to be fairly constant and equals ~
0.016 (mass ratio: 0.010). This value is consistent with other studies such as from satellite measurements (OCO-
2/3 and TROPOMI) and the bottom-up inventories reported by MacDonald et al. (2023). Finally, we estimated the
CO emissions using the average daily growth rate determined from measurements at the UNA station. Although
this method likely leads to an under-estimate of the emissions due to the non-negligible effects to the advection,
our results were found to be very consistent with the 2016 and 2018 SEDEMA inventories. The same strategy
could not be applied at the VAL station likely because of the dominant and rapid transport of the air masses before
noon, preventing them from accumulating. The UNA station is located near a topographic barrier (the
Chichinautzin mountain range), which favours regional turbulence and rapid homogenization of the air masses,
making the total column measurements representative of the emissions. We finally estimated the $CO_2$ emissions
using the CO growth rate and the $CO/CO_2$ ratio. The finding that our $CO_2$ emission estimates are within 20% of
those of SEDEMA for total emissions show that our ratio reflects not only the traffic sources but is also affected
by other sources such as industrial activities and domestic burning. The UNA station, with its advantageous
orography, is therefore a good site to capture well-mixed emissions from the city and serves as a site to follow the
interannual variability and trends of the emissions in this urban environment. Finally, this study showed the
feasibility to monitor the long-term evolution of anthropogenic $CO_2$ and CO emissions in Mexico City by
deploying only a few EM27/SUN instruments.

## 7 Author contribution

All the co-authors contributed in the discussion of concepts, and to the preparation of the manuscript. NT, WS and
MG were responsible of FTIR measurements and the data analysis. MG and WS lead the ALTZ station
development and its long-term operation. AB and EGC were responsible of the maintenance of the instruments at
the Altzomoni station. VA helped to classify the days and hours with low ventilation and strong turbulence and



provided the UNAM emission inventory. EGC was in charge of the in-situ measurements, with the support of OL.
MG and MR led the MERCI-CO2 project. FH lead at KIT the German-Mexican collaboration for the deployment
of the high resolution FTIR spectrometer and supports its long-term operation as part of NDACC. FH has helped
in the design and setup of the spectrometer and solar tracker before it was shipped to Mexico. He has developed
the retrieval code PROFFIT and gives continuously support to the UNAM group for its use and in operating the
spectrometer. FH and CA lead the German-Mexican collaboration and give precious help for the EM27/Sun
measurements in the frame of the COCCON network. All the co-authors contributed of the redaction of the
manuscript.
**8 Competing interests**
The authors declare that they have no conflict of interest.
**9 Acknowledgements**
We acknowledge the CONACyT-ANR project 290589 'Mexico City's Regional Carbon Impacts' (ANR-17-
CE04-0013-01) for funding. Also the former projects CONACYT 239618 "El estudio del ciclo de Carbono y de
los gases de efecto invernadero utilizando espectroscopia de absorción solar" and UNAM-DGAPA
PAPIIT IN111521/IN106024 are acknowledged. We acknowledge the CONACyT-ANR project 290589 'Mexico
City's Regional Carbon Impacts' (ANR-17-CE04-0013-01) for funding. We acknowledge the technical assistance
provided by Omar López, Alfredo Rodriguez, Miguel Robles, Delibes Flores, and the Instituto de Ciencias de la
Atmósfera y del Cambio Climatico (UNAM) for the institutional support to carry out this study. We thank T.
Blumenstock from the KIT for his precious help and fruitful discussion during the last years. We thank Dr. Thomas
Boulesteix for his help at the Altzomoni site, his fruitful discussions.

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
