# Peer review of "CO2 and CO temporal variability over Mexico City from ground-based total column and surface measurements"

_EGUsphere, 2024_

## Author Comment (AC1)

**Response to Referees**

*Atmospheric Chemistry and Physics*

*Taquet et al.:* *"*$CO_2$ and CO temporal variability over Mexico City from ground-based total column and surface measurements*"*

We thank the two reviewers for their very constructive comments, which helped to prepare an improved revised manuscript.

**1.    Response to Reviewer #1**

**Overview:**

The paper by Taquet el al reports on measurements from Mexico City, including three sites, two within the city itself and one at the high-altitude site at Altzomoni. This group is very experienced in FTIR columns measurements, using both high resolution FTIR spectrometers at fixed locations as part of the NDACC/TCCON networks, but also the use of portable EM27/SUN FTIR spectrometers in the COCCON network. This study uses quite an extensive set of data and a methodology based on previous work reported by Stremme et al in 2013. The data set and instrumentation are described in good detail, and any reader who wishes to understand the exact procedure will need to refer to the Stremme paper for details. It is my impression that what the authors produce here is a very promising report of Mexico emissions over a 5-year period and are able to compare this with in situ data as well as satellite based TROPOMI measurement. Within the constraints of their method, that is a simplified approach that avoids the use of complicated modelling, these data sets seem to compare well, given the spatial and inherent uncertainty limitations of the ground-based column measurements.

In terms of principal criteria, the manuscript is rated as good (3) for scientific significance (the methods are not new but are followed with a highly experienced team), rated excellent (4) for scientific quality, an in general rated as good(3) to excellent(4) in presentation quality (see technical corrections below)

In general, the manuscript is well written, gives a very good scientific motivation for the work, has clear description of the measurements and references the paper by Stremme where a very detailed account of the method can be read (not required to be repeated here).

**Comments:**

Given that understanding of emissions in mega cities is a global question, is the study method only limited to Mexico City? Could this method be applied to others cities or does Mexico city offer very unique geography that means this cannot really be applied elsewhere?

**Reply:** We thank the reviewer for highlighting this aspect. Indeed, the simple methodology employed in our manuscript for estimating emissions holds potential applicability to cities experiencing periods of reduced ventilation during some hours of the day and therefore it mainly depends on the geographical location and topography. For cities characterized by flat topography, the wind might play a key role and correlation between wind direction/speed and the carbon monoxide column might be important for most approaches.

There are already plenty of approaches for satellite-based column measurements like Pommier et al. (2013) and Tu et al. (2020), which map the measurements upwind and downwind the source to estimate the emissions. Conceptually, a similar approach could be applied using two or more ground-based instruments (Hase et al., 2015: Chen et al., 2016; Frey et al., 2018) positioned upwind and downwind of the urban plumes. However, the efficacy of such methodologies is contingent upon meteorological conditions, often limiting emission estimates to short-term periods and constraining statistical analyses. This is the case in our study during which the UNA station was dominantly downwind (see Figure S6) under the prevailing wind conditions, and VAL never is a representative background station because of its situation in a highly industrialized area. Our methodology has so far only been applied to Mexico City (Stremme et al., 2013), which presents a special combination of low daytime ventilation conditions and an urban area large enough for the column growth rate to be dominated by the emission flux. Most cities fall somewhere in between the limits where the growth rate of the column is directly related to emission fluxes and the downwind-upwind difference is an important key to get the emissions. However, it is likely that a similar simple method, tailored to the unique geography of each city, might be applicable in many cases.

In our manuscript we added at l. 875:

"The methodology employed here for monitoring the long-term temporal variability of CO emission fluxes is likely to be adapted to other urban areas where the topography damps the ventilation down for several hours each day, thereby establishing that the column growth rate is dominated by the emission flux."

Why not use a chem/trans model in this study? Clearly there is the yet to be published study by Che et al presumably on a subset of this data. Are there specific reasons why a more complicated modelling exercise is not undertaken for the entire measurement record by these authors? Is the suggestion that this simpler approach here should be adopted elsewhere?

**Reply:** A chemical/transport model provides valuable information to track cities emissions and understand the source processes using either surface or columnar measurements. Che et al. (2024) and Che et al. (submitted to *Journal of Geophysical Research*) successfully estimate Mexico city $CO_2$ emissions using the data of the MERCI-CO2 intensive campaign (Oct2020 - May 2021) and obtains very consistent results with inventories using both space- and ground-based measurements and the lagrangian XSTILT model.

Nevertheless, such models necessitate substantial computational time and memory resources and a large number of measurement stations to ensure statistical robustness.

Such models require some approximations due to the available data (i.e., meteorological fields, prior information, background estimate, etc.), which, while partially representative of the complexity of atmospheric turbulence and mixing at fine spatial and temporal scales, are not exhaustive. The optimization of the model configuration, which relies on specific parametrizations, can be not straightforward, in particular for long-term periods and, particularly in regions with complex orography. Despite numerous recent efforts to quantify the different types of error due to these approximations, it remains a challenge to ascertain to what extent the results depend on the assumptions and parameters used.

Our study highlights that at specific stations and under specific climatic and topographic conditions, columnar measurements and their growth rate are directly connected to emissions. The emission data derived directly from columnar measurements provide independent information from the transport/chemistry models. This data can be used not only to explore the temporal variability of the emissions of a city but also to identify possible inaccuracies in the parameterization of complex models. Nevertheless, it is evident that our approach cannot supplant more sophisticated chemical transport models in furnishing more precise data or absolute values concerning emissions, chemical transport, and sources.

We thank the reviewer for raising this point and add at the end of the manuscript at l.878:

"Although the straightforward model presented here is not intended to replace a complex transport/chemical model for a precise estimate of city emissions, the results obtained demonstrate that it is nevertheless possible to track their temporal evolution with a high degree of reliability."

A few other comments;

1. Page 19, line 561: are these low costs sensors different to the CO2 sensor mentioned on page 7?

   **Reply:** We thank the reviewer for mentioning this unclear point:

   The low cost sensors mentioned at this line of the manuscript are the same as the one referred to in page 7. The following description was added to Page 7 of the new manuscript, line 220:

   "Additionally, the VAL site included a low-cost medium precision $CO_2$ sensor, as a part of a network implemented during the MERCI-CO2 campaign. It consists of a NDIR-type of sensor (SenseAir, model HPP3) that can measure in the 0 to 1000 ppm range and after a calibration and target gas follow-up procedure, can produce data with <1% accuracy (Porras et al., 2023)." We also added the reference Porras et al. (2023) at line 566.

2. Page 21, lines 619/620: what is the significance of these slight decreases?

   **Reply:** The slight decrease observed in the $\Delta$XCO atmospheric concentration trend (which is consistent with the surface CO decreasing trend) was also reported in other longer-term studies (Garcia-Franco, et al., 2019; Molina, 2021, Hernández-Paniagua et al., 2021). The decreasing trend likely results from the successive air quality management programs implemented in the CDMX since the 1990s to improve the air quality, which combined regulatory actions with technological change based on scientific, technical, social, and political considerations (Molina, 2021). Vehicle emissions were curbed through technological advancements and fuel quality enhancements (including removal of lead from gasoline, mandatory use of catalytic converters, reinforcement of vehicle inspection and maintenance, mandatory "no driving day" rule), while industrial and commercial emissions were mitigated by measures such as refinery closures, industrial relocation, fuel substitution, ect. The decreasing trend is also accentuated by including in the analysis the COVID-19 lock-down period, for which the monthly variability and average significantly decreased compared to the previous years. The slight decrease also observed for $\Delta XCO_2$ in Figure 8 likely reflects the same facts, given that an important part of anthropogenic $CO_2$ emissions in Mexico are due to the mobile sources.

We added in the manuscript in l.639:

"The long term $\Delta_m XCO$, also observed in other studies (Garcia-Franco, et al., 2020; Molina, 2021, Hernández-Paniagua et al., 2021) likely reflect the successive air quality management programs implemented in the CDMX since the 1990s to improve the air quality, including technological advancements and fuel quality enhancements as well as refinery closures, industrial relocation, or fuel substitution."

3. Page 21, line 622: possible reasons for the low ratios?

**Reply:** The observation of lower $\Delta XCO/\Delta XCO_2$ ratios during the raining season (where both CO and $\Delta XCO_2$ are minimum) are in accordance with the observation of Linian-Abanto et al. (2021). This seasonal dependence could be the result of (1) a change in the relative contribution of the different types of sources measured at the stations driven by a change in the dominant wind direction (2) a change in the turbulence/mixing conditions and pollutant concentration driven by the meteorological synoptic patterns.

(1) Typically low emission ratios ($CO/CO_2 < 0.02$) correspond to high combustion efficiencies, originating from the burning of well-processed liquid fuels or gasses (vehicle engines, natural gas stoves, etc.) while higher emission ratios ($CO/CO_2$ from 0.03 to 0.1) reflect low combustion efficiency, due to use poorly processed solid fuels (coal stoves or biofuels, biomass burning, etc.) (Liñán-Abanto et al., 2021 and therein references). Therefore higher $\Delta XCO/\Delta XCO_2$ are expected during the typical period of the biomass burning contrasting with the rest of the year.

(2) Since this seasonal occurrence of low ratio during the rain season is observed at both VAL and CCA sites, it is likely explained by synoptic meteorological patterns. Typically, within the MCMA, the most severe pollution episodes happen in winter (cold dry season) and spring (warm dry season) due to the formation of strong surface-based inversions overnight and early in the morning.

These inversions create highly stratified atmospheric conditions that trap vehicle emissions and industrial pollutants near the surface. Late spring is the season when ozone concentrations in México City often surpass the normative limits and the government takes action to reduce emissions from the traffic sector. Conversely, in summer, a deep easterly flow over Mexico City brings abundant tropical moisture from the Gulf of Mexico, resulting in frequent cloud cover and rainfall. This weather pattern reduces the occurrence and intensity of nocturnal inversions and aids in washing away pollution. Therefore, the lower $\Delta XCO/\Delta XCO_2$ levels observed during the rainy season are likely due to a reduced contribution of polluted air masses originating from the city.

We added in line 643: "Regarding the low seasonal variability observed for the $CO/CO_2$ ratios, it is likely related to mass burning episodes and high-pressure weather conditions that occur during the dry season."

4. Page 22, line 654: are there other independent traffic patterns that might shed light on the lack of lock-down signal?

**Reply:** A difference in the $\Delta XCO_2$ and $\Delta XCO$ (UNA-VAL) was expected during the COVID-19 lock-down period, especially for CO, because the main source of CO near UNAM is the road traffic, in contrast with the Northern part of the city, which is highly industrialized. Due to the suspension of the academic activity, reduced business activity and remote work becoming widespread during the COVID19 lock-down period, the reduction of the road traffic in this area was significant (Hernández-Paniagua et al., 2021: Supplementary Material Table S1).

The absence of lower $\Delta CO$ (UNA-VAL) values in Figure 10 shows either that the decrease in the CO emissions is homogeneous across the city or that some other phenomenon is masking the local decrease in these emissions. In the study by Hernández-Paniagua et al. (2021), which examines the impact of the COVID-19 lockdown on the MCMA air quality, the effect of the lockdown is clearly observed regarding the $NO_2$ (a robust tracer of motor vehicle emissions) variability but much less evident for CO. They first highlight the role of the meteorological conditions (accumulation of contaminants during stagnant atmospheric conditions which can mask their temporal variability) and attribute the quasi absence of the CO anomaly to a possible increase of the domestic liquid petroleum and natural gas burning because of the stay-at-home order.

Figure 3 of our study shows a negative anomaly during these months at both VAL and UNA stations, showing that the long-term total column variability captured a decrease of the CO emissions due to the global lock-down effect at the two stations. However, the fact that no difference is observed in horizontal (VAL-UNAM) $\Delta CO$ gradients shows that the two stations captured the composition of a homogeneously mixed layer, which may be due to the stagnant atmospheric conditions that tend to favor the accumulation of pollutants (i.e: on an intraday scale), and mask the impact of local sources.

**Technical Corrections:**

1.  Page 6, line 179: Nation -> Nafion Done
2.  Page 7, line 215: what is the CO2 sensor (ie, type, model etc)?

    **Reply:** See response to the comment 1

3.  Page 9, line 279: remove "the" Done
4.  Page 9, line 295: remove "the" Done
5.  Page 10, line 324: "in order of" probably sounds better with "of order of" Done
6.  Page 18, line 553: "the total columns XCO2 and XCO" –> "the total column mole fractions XCO2 and XCO…" We replace this part by "The $XCO_2$ and XCO" to simplify the text, the XCO2 and XCO being defined before.
7.  Page 24, line 684: " upwind the city…" -> "upwind of the city…" Done
8.  Page 24, line 689: " ..in the Stemme…" -> " ..in Stremme …" Done
9.  Page 24, line 698: "… mountain around …" -> " … mountains around..." Done
10. Page 24, line 709: "… would be .." -> " …is…" Done
11. Page 24, line 711-714: does it matter though if these uncertainties have both systematic and random components?

    **Reply:** We agree with the reviewer that the relative contribution of the random and systematic error is not reported in our manuscript. As we aim to estimate the average emissions, we only use one "extrapolation factor" in time and space. By definition, random error can be reduced by $\sqrt{(N-1)}$ when averaging N measurements, while systematic error can't.
    The distinction between random and systematic errors would only make sense if the time extrapolation factor can be based on traffic activity measurements and the spatial extrapolation factor can be derived from a statistically representative number of distributions on individual days. We did not use this strategy in our study because we only present a rough estimate of the uncertainty.

    Anyway, Stremme et al. (2013) report an evaluation of the error due to the temporal extrapolation, using the modeled temporal distribution at various times (boxcar, triangle, trapezoid distributions and the distribution from the official inventory) and found a standard deviation (STD) of 26 % × AVG and a standard error of ≈ 10 %= 26 %/$\sqrt{(N-1)}$ assuming N=8 distributions. As the result is an average estimate of the emission and many days contribute to the estimation, the systematic error due to the temporal interpolation factor is likely much higher than the random error of the fitted growth rate.

12. Page 24, line 717: what are these instrumental and retrieval effects, just briefly? What size are these factors?

    **Reply:** We more specifically refer to the airmass dependent effect mostly affecting the $CO_2$ due to spectroscopic inadequacies (e.g. line widths, neglect of line-mixing, inconsistencies in the relative strengths of weak and strong lines). This effect can affect the intraday pattern of $CO_2$ (Wunch et al., 2010), if the actual profile of the target gas in the atmosphere differs from the a-priori profile assumed in the retrieval. PROFFAST applies an Airmass Dependent

Correction Factor (ADCF) similar to TCCON (Deutscher et al., 2010; Wunch et al., 2010) but this effect is not yet fully resolved and can cause some imprecisions in the diurnal patterns. To give an idea of the influence of this effect we used the ADCF equation and the coefficient reported in the technical note of the COCCON website (https://www.imk-asf.kit.edu/downloads/Coccon/2021-04-30_Instrument-Calibration.pdf) corresponding to the used version of PROFFAST:

$$Xcorr\_adcf(x)/Xuncorr\_adcf = \{1 + x^4 \cdot (b + c \cdot x^8)\} / \{1 + x^4_{ref} \cdot (b + c \cdot x^8_{ref})\}$$

where b and c are the ACDF coefficients. We calculate the correction for the minimum and maximum SZA compliant with our applied filters (SZA<70). For the SZA close to zero the correction is minimum (ADCF close to 1.). The relative difference between the two results was found to be 0.18% for $CO_2$ (which corresponds to about 0.7 ppm) and 3.8% for CO (which corresponds to about 0.005ppm). Therefore the correction can be significant for diurnal variability of $CO_2$ (<1-2 ppm) while it can be neglected for studying CO anomalies (>0.02ppm). A further retrieval-associated effect one might discuss here is the non-ideal column sensitivity of the retrieval. It seems, for CO in the PBL and assuming small SZA, it is near 0.95 (so only 5% underestimation), and for CO2 it is ~ 1.25, so ~ 25% overestimation in the retrieval.
We complement the following lines (l. 738-740) in the manuscript:
"$CO_2$ emissions could not be directly estimated using the same method, given its complex diurnal pattern, which is a cumulative result of both natural and anthropogenic contributions and likely been influenced by additional factors, related to instrumental and retrieval effects (i.e: airmass dependence error with a sub-percentage error for $CO_2$, non-ideal column sensitivity of the retrieval which represent near 25% overestimation for $CO_2$ anomaly and 5% underestimation for CO anomaly in the PBL.)"

13. Page 25, figure 11 caption, line 731: is that t/year or kt/year? Done
14. Page 27, lines 807/808: The mention of other components is presumably industrial and domestic burning as described in the next sentence? Need to link these two sentences more clearly.
    **Reply:** We agree with the reviewer that this part was unclear and rephrased the two sentences in l. 828-836:
    "The CO/$CO_2$ ratios calculated from the SEDEMA data for total emissions are similar to ours (0.014 and 0.011 in 2016 and 2018, respectively), suggesting that our average CO/$CO_2$ ratio is actually representative of the global mixing of the different sources of the MCMA, and not only dominated by the road traffic. Interestingly, according to the SEDEMA inventory, road traffic, the main anthropogenic CO source is identified by ratios (0.019 and 0.016 in 2016 and 2018, respectively) only slightly higher than our global average; whilst the industrial and domestic burning sectors, which represent the second main $CO_2$ anthropogenic sources, produces a one order of magnitude lower ratio. In any case, our measurements are well representative of the main source of the CO and $CO_2$ anthropogenic emissions".

15. Page 28, line 837: ".. effects to the advection, …" -> " ..effects of advection, .." Done

16. Page 29, line 861: "redaction" is not the correct term which means to remove text for publication, so "writing" is better here. Done

**2. Response to Reviewer #2**

Taquet et al. investigated the variability of $CO_2$ and CO in the Mexico City Metropolitan Area (MCMA) on different (annual, seasonal, and diurnal) time scales, based on ground-based in situ and remote sensing measurements. Enhancement ratios ($CO/CO_2$) were derived from both the in situ and remote sensing measurements and used to estimate $CO_2$ emissions in the MCMA by combining them with TROPOMI CO data. The estimated annual $CO_2$ emissions showed the reduction in 2020, likely due to the COVID-19 lockdown, which in not yet reflected in the emission inventories.

The topic of this manuscript is important and relevant to the scope of Atmospheric Chemistry and Physics. In addition, the analysis method is appropriate, and the writing structure is well organized. I recommend that this article be published after addressing the following concerns and questions.

Specific comments

Abstract: The abstract only describes what was done in this study, so please write what was revealed.

**Reply:** We replaced some parts of the abstract to highlight our findings:

"Accurate estimates of greenhouse gas emissions and sinks are critical for understanding the carbon cycle and identifying key drivers of anthropogenic climate change. In this study, we investigate the variability of CO and $CO_2$ concentrations and their ratio over the Mexico City Metropolitan Area (MCMA) from long-term time-resolved columnar measurements at three stations, using solar absorption Fourier transform infrared spectroscopy (FTIR). Using a simple model and the mixed layer height from a ceilometer, we determined the CO and $CO_2$ concentration in the mixed layer from the total column measurements and found good agreement with surface cavity ring-down spectroscopy measurements. In addition, we used the diurnal pattern of CO columnar measurements at specific time intervals to estimate an average growth rate that, when combined with the space-based TROPOMI CO measurements, allowed deriving annual CO and $CO_2$ MCMA emissions from 2016 to 2021. A decrease of more than 50% of the CO emissions was found during the COVID19 lockdown period with respect to the year 2018. These results demonstrate the feasibility of using long-term EM27/Sun column measurements to monitor the annual variability of anthropogenic $CO_2$ and CO emissions in Mexico City without recourse to complex transport models. This simple methodology could be adapted to other urban areas if the orography allows low ventilation for several hours per day, which allows that column growth rate to be dominated by emission flux."

L97: What does the "ground-based satellite produce" mean?

**Reply:** We thank the reviewer to detect this mistake, and replace this part by "atmospheric monitoring and satellite products validation"

L129-131: Please add latitude, longitude, and elevations of the VAL, UNA, and ALTZ stations. Done

Figure 1: What do the triangle and cross symbols represent?

**Reply:** We thank the reviewer for pointing out that the information was missing. We added the information in the legend of Figure 1 and complement the manuscript in l. 707 and l.721-722.

L179 and L210: Nation air dryer → Nafion air dryer Done

L276: VRM-scaling → VMR-scaling Done

L281: The degree of freedom for the CO retrievals in the MIR region is not expected to be as large. Do you evaluated the impact of using a single prior in the profile retrieval of CO?

The CO retrieval from the MIR measurements is a Network for Detection of Composition Change (NDACC) product (Pougatchev et al., 1994; Rinsland et al. 1998) for more than 30 years. There are typically up to 4 degrees of freedom of signal (DOFs) in the profile retrieval, with information from the bottom up to the upper stratosphere (Velasco et al. 2007, Borsdorff et al 2014). High mountain sites might have less DOFs. In Mexico City, we have a lower spectral resolution and in Altzomoni, due to the altitude above 4000 m, we also have slightly less degrees of freedom. In the NDACC retrievals, a fixed a priori is normally used, so that the measured change is coming from the measurements and not from variable a priori information. The "block constraint", as described by v.Clarmann and Grabowsky, (2007) ensures that the growth rate in the mixing layer is not damped and the impact of the free troposphere to the column in the mixed layer is very small. It is important to use a single apriori, to be sure that the column growth rate is a result of the measurement and not introduced by a variable a-priori information.

We added the citations in l. 285: "(Pougatchev et al., 1994; Rinsland et al. 1998)"

L419: The description of Figure 4 in the text precedes Figure 3. Please swap the order of Figures 3 and 4. Done

Figure 3: Figures 3C and 3D are not explained in the text. Please add their explanations or omit these figures. Done: We added references of these figures in the text in l. 481, 482, 484 and 499.

L479-480: To understand what is described in this sentence, which figure should readers refer to? Done: We cite the figure 4C in this sentence.

Figure 5: What factors contribute to the difference in the diurnal patterns in $\Delta XCO_2$ between the UNA and VAL sites? Can this difference be explained by differences in the spatiotemporal patterns of wind direction within the MCMA?

We really appreciate the commentary of the reviewer and detail below the possible reasons which can explain this difference.

In our manuscript, the XCO and $XCO_2$ diurnal patterns were calculated after discarding days with high ventilation, based on ERA5 data. Figure S6 shows wind rose diagrams characterizing the surface wind (at 10 m) measured by the local UNA and VAL meteorological stations using the RUOA and REDMET networks, after selecting the data which comply with the Ventilation Index filter.

[Figure]

*Figure S6: Dominant surface wind speed and direction at the UNA and VAL stations (average over 01/09/2019-01/06/2021) calculated from the REDMET(VAL) and RUOA (UNA) meteorological stations, selecting days complying with the VI filter described in the manuscript.*

Figure S6 shows a dominant average surface wind direction over the Mexico valley from the North, at least after 10 LT. However, a real difference appears in terms of spatio-temporal variability at the scale of the MCMA. While the wind rose diagram for the UNA station shows an important disparity in surface wind direction and a significant intraday variability, the VAL station shows a very constant wind direction all day long, which mainly coincide with the CO distribution observed from the Tropomi data in Figure 1 of the manuscript. The wind direction and advection of the airmass near the VAL station are likely mainly controlled by the topographic barriers of the region see topography in the new version of Figure 1), which can explain the gradient in the CO distribution upwind of the VAL station. The airmass measured at the VAL station likely has a contribution of both local sources emissions and airmass coming from the north. In contrast, near the UNA station, the flat ground allows a more efficient mixing and due to the dominant North-NorthEast wind component, the captured airmass likely often reflects the MCMA plume emissions. In addition the West-Northwest wind component at UNA is likely to be the effect of down-slope flows from the mountain ridge in the early morning (6 - 9 LT). At VAL, the plateau-to-basin winds are the main influx into the basin coming from the northwest in the morning. There can also be an influence from an up-valley flow in the mornings (de Foy et al., 2006). These observations are supported by the carbon monoxide distribution over the MCMA from Tropomi, shown in Figure 1 of the manuscript. The gradients in the total columns of carbon monoxide shown by Tropomi (Figure 1) are different near the VAL and UNA stations, and even the same global ventilation pattern would impact both sites differently. Especially the stronger gradient in the typical upwind direction will lead to a high variability at VAL. Only at UNAM the area is homogenous enough, so that we can assume that the ventilation plays a minor role during morning and up to noon.

We modified Figure 1 to highlight the topography of the region. We added the figure S6 in the supplementary data and added the following lines in the manuscript l. 584-594:

"The difference observed between the diurnal pattern of the XCO and $XCO_2$ at VAL and UNA is likely due to the different advection drivers in the region mainly controlled by the topography. A Northern surface wind direction (Figure S6) is generally dominating over the Mexican valley but is locally highly influenced by the mountainous barriers. The West-northwest wind component at UNA is likely to be the effect of down-slope flows from the mountain ridge in the early morning (6 – 9 LT mostly), while at VAL, the plateau-to-basin winds are the main influx into the basin coming from the northwest in the morning. There can also be an influence from an up-valley flow in the mornings (de Foy et al., 2006). More generally the VAL station is likely influenced by the north mountain, generating a significant gradient in the CO distribution upwind of the VAL station (Figure 1). In contrast, near the UNA station, the flat ground allows a more efficient mixing and due to the dominant North-Northeast wind component in the late morning, the captured airmasses likely often reflects the MCMA plume emissions."

and at l.864: "The same strategy could not be applied at the VAL station, likely because of dominant southward advection of the airmass, due to the complex topography in this part of the MCMA. In contrast, the UNA station is located in a flat ground downwind of the main anthropogenic source of the MCMA which likely allows establishing a direct relationship between the columnar measurements and the MCMA CO and $CO_2$ emissions."

L529 and L798: Please define the "MGRA". We thank the reviewer for the types here and replace "MGRA" by "MAGR"

L530: What does the "ELD" represent? Please add the explanation.

**Reply:** We added some explanation in lines 533-536: "To explore the 2020 lock-down influence on the diurnal pattern, three different periods were distinguished for each plot, the first one (blue trace: 2016 - 2021) corresponding to the whole measurement period excluding the interval between March and June 2020 corresponding to the lock-down period (hereafter, called "ELD" for "excluding the lock down period"), where a significant MAGR decrease was observed;"

Figures 7A and 7B: Are the "Surface" and "From FTIR Tot.col." legends reversed?

**Reply:** We thank the reviewer for pointing out this mistake and modified the legend.

L648: Fig. 9 instead of Fig. 7?

**Reply:** We thank the reviewer for pointing out this mistake and replaced "Fig.7" by "Fig. 9".

L683: Over what domain is $(CO_{MCMA} - CO_{bgrd})$ integrated? Area?

**Reply:** We added in l. 703: "In Eq. (8), $(CO_{MCMA} - CO_{bgrd})$ is integrated over the area where the CO TROPOMI total columns are higher than a predefined background value." The way to define the background value is explained in the following sentences.

L694: What does the "mixed layer column" mean and how is it defined?

**Reply:** We thank the reviewer for pointing out this unclear point and we replaced the sentence at l. 714: "The mixed layer column at UNA from the TROPOMI data was found to be $1.93 \times 10^{18}$ molec.cm$^{-2}$ (Fig. 1), which is consistent with our EM27/SUN ground-based measurements (average of $2.17 \times 10^{18}$ molec.cm$^{-2}$)" with:

"The fresh CO was estimated from the TROPOMI data by removing the background ($1.45 \times 10^{18}$ molec.cm$^{-2}$) to the average total columns found at UNA ($1.93 \times 10^{18}$ molec.cm$^{-2}$) and was found to be $4.79 \times 10^{17}$ molec.cm$^{-2}$."

L717: The factors related to instrumental and retrieval effects would also affect the CO columns. How do these factors affect the CO emission estimates?

**Reply:** We more specifically refer to the airmass dependent effect mostly affecting the $CO_2$ due to spectroscopic inadequacies (e.g. line widths, neglect of line-mixing, inconsistencies in the relative strengths of weak and strong lines). This effect can affect the intraday pattern of $CO_2$ (Wunch et al., 2010), if the actual profile of the target gas in the atmosphere differs from the a-priori profile assumed in the retrieval. PROFFAST applies an Airmass Dependent Correction Factor (ADCF) similar to TCCON (Deutscher et al., 2010; Wunch et al., 2010) but this effect is not yet fully resolved and can cause some imprecisions in the diurnal patterns. To give an idea of the influence of this effect we used the ADCF equation and the coefficient reported in the technical note of the COCCON website (https://www.imk-asf.kit.edu/downloads/Coccon/2021-04-30_Instrument-Calibration.pdf) corresponding to the used version of PROFFAST:

$$Xcorr\_adcf(x)/Xuncorr\_adcf = \{1 + x^4 \cdot (b + c \cdot x^8)\} / \{1 + x^4_{ref} \cdot (b + c \cdot x^8_{ref})\}$$

where b and c are the ACDF coefficients.

We calculate the correction for the minimum and maximum SZA compliant with our applied filters (SZA<70). For the SZA close to zero the correction is minimum (ADCF close to 1.). The relative difference between the two results was found to be 0.18% for $CO_2$ (which corresponds to about 0.7 ppm) and 3.8% for CO (which corresponds to about 0.005ppm). Therefore the correction can be significant for diurnal variability of $CO_2$ (<1-2 ppm) while it can be neglected for studying CO anomalies (>0.02ppm). A further retrieval-associated effect one might discuss here is the non-ideal column sensitivity of the retrieval. It seems, for CO in the PBL and assuming small SZA, it is near 0.95 (so only 5% underestimation), and for CO2 it is ~ 1.25, so ~ 25% overestimation in the retrieval.

We complement the following lines (l. 738-740) in the manuscript:

"$CO_2$ emissions could not be directly estimated using the same method, given its complex diurnal pattern, which is a cumulative result of both natural and anthropogenic contributions and likely been influenced by additional factors, related to instrumental and retrieval effects (i.e: airmass dependence error with a sub-percentage error for $CO_2$, non-ideal column sensitivity of the retrieval which represent near 25% overestimation for $CO_2$ and 5% underestimation for CO in the PBL)".

L758: Please define the "GRA". We replace "GRA" with "AGR".

Supplementary file

Caption of Figure S2: after aplying the calibration factors → after applying the calibration factors Done

Table S1: Are the digits of the calibration factor of "VERTEX-XCO MIR" insufficient? We added 2 additional digits for the calibration factor.

Caption of Table S3: *corresponds → The asterisks (*) correspond Done

[revised manuscript text omitted]

[1] only includes the propagated growth rate error. An estimation of errors due to the spatial and temporal interpolation is given in Figure 11 and discussed in the text.